

# Biogeochemistry of climate driven shifts in Southern Ocean primary producers

Ben J. Fisher[1], Alex J. Poulton[2], Michael P. Meredith[3], Kimberlee Baldry[4], Oscar Schofield[5], Sian F. Henley[1].

[1] School of GeoSciences, University of Edinburgh, Edinburgh, United Kingdom
[2] The Lyell Centre for Earth and Marine Science, Heriot-Watt University, Edinburgh, United Kingdom
[3] British Antarctic Survey, Cambridge, United Kingdom
[4] Institute for Marine and Antarctic Studies, College of Sciences and Engineering, University of Tasmania, Hobart, TAS, Australia
[5] Center of Ocean Observing Leadership, School of Environmental and Biological Sciences, Rutgers University, New Brunswick, NJ 08901, USA

*Correspondence to*: Ben J. Fisher (ben.fisher@ed.ac.uk)

**Abstract.** As a net source of nutrients fuelling global primary production, changes in Southern Ocean productivity are expected to influence biological carbon storage across the global ocean. Following a high emission, low mitigation pathway (SSP5-8.5), we show that primary productivity in the Southern Ocean is predicted to increase by up to 30% over the 21st century. The ecophysiological response of marine phytoplankton experiencing climate change will be a key determinant in understanding the impact of Southern Ocean productivity shifts on the carbon cycle. Yet, phytoplankton ecophysiology is poorly represented in Coupled Model Intercomparison 6 (CMIP6) climate models, leading to substantial uncertainty in the representation of their role in carbon sequestration. Here we synthesise the existing spatial and temporal projections of Southern Ocean productivity from CMIP6 models, separated by phytoplankton functional type, and identify key processes where greater observational data coverage can help to improve future model performance. We find substantial variability between models in projections of light concentration (>15000 $(\mu E \ m^2 \ s^{-1})^2$) across much of the iron and light limited Antarctic zone. Projections of iron and light limitation of phytoplankton vary by up to 10 % across latitudinal zones, while the greatest increases in productivity occurs close to the coast. Temperature, pH and nutrients are less spatially variable, projections for 2090-2100 under SSP5-8.5 show zonally averaged changes of +1.6 °C, -0.45 pH units and Si* decreases by 8.5 $\mu mol \ L^{-1}$. Diatoms and pico/misc phytoplankton are equally responsible for driving productivity increases across the Subantarctic and Transitional zones, but pico and misc phytoplankton increase at a greater rate than diatoms in the Antarctic zone. Despite the variability in productivity with different phytoplankton types, we show that the most advanced models disagree on the ecological mechanisms behind these productivity changes. We propose that a sampling approach targeting the regions with the greatest rates of climate-driven change in ocean biogeochemistry and community assemblages would help to resolve the empirical principles underlying phytoplankton community structure in the Southern Ocean.



## 1. Introduction

The biological uptake of carbon by marine phytoplankton represents an important process in the Earth system (Deppeler and

Davidson, 2017), with ocean carbon storage mediating atmospheric $CO_2$ concentrations, including $CO_2$ of anthropogenic origin (Riebesell et al., 2007). Across the global ocean, uptake of carbon accounts for ~25% of $CO_2$ released by human activities (Friedlingstein et al., 2022). The Southern Ocean is a disproportionately large carbon and heat sink relative to its size (Frölicher et al., 2015), accounting for 30-40% of this global anthropogenic $CO_2$ uptake (e.g. Caldeira and Duffy, 2000;DeVries, 2014), predominantly due to enhanced atmosphere-ocean exchange at increased atmospheric $CO_2$ concentrations (Friedlingstein et

al., 2022). While biological uptake is considered to play a minor role in total $CO_2$ uptake (Landschützer et al., 2015;Gruber et al., 2019), variability in $pCO_2$ has been associated with summertime blooms in the Southern Ocean (Gregor et al., 2018;Coggins et al., 2023). Under a future climate scenario with longer growth seasons (Moreau et al., 2015), increased seasonal productivity (Leung et al., 2015;Fu et al., 2016) and a reduction in ocean $CO_2$ absorption efficiency (higher Revelle factor) (Hauck et al., 2015); biological and physical drivers of carbon exchange across the air-sea interface are likely to undergo

substantial changes. As the Southern Ocean's ability to buffer increased concentrations of atmospheric $CO_2$ weakens, the role of pelagic ecosystems, are expected to become increasingly important in Southern Ocean carbon uptake (Henley et al., 2020).

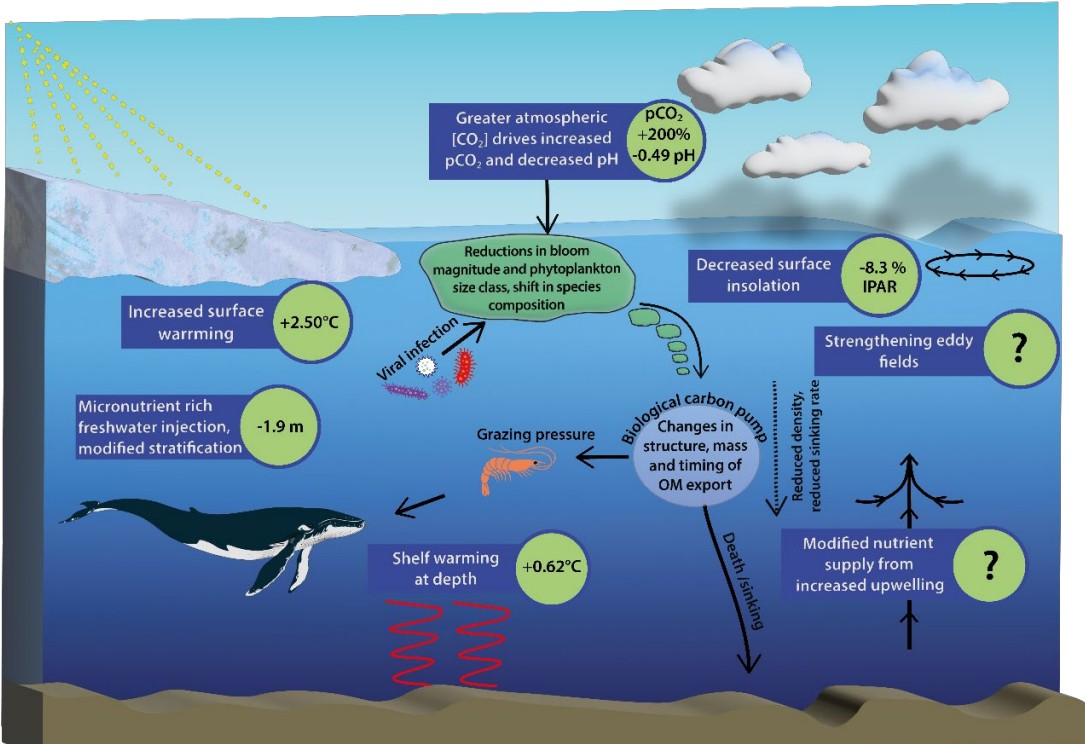

**Figure 1: Schematic diagram of Southern Ocean pressures associated with climate change and the downstream biogeochemical consequences for ecosystem productivity. Values shown are 100 year mean changes to 2100 under the SSP5-8.5 scenario south of 65°S and are taken from CMIP6 models and existing literature (McNeil and Matear, 2008;Purich and England, 2021)  (see Table S1 for a full description). Question marks indicate processes where estimations of change do not currently exist.**





Small celled marine phytoplankton (0.002-0.2 mm) are responsible for the production of biological carbon, fuelling ecosystems
across the Southern Ocean, but are vulnerable to environmental change because of their specific requirements for light and
iron, which are the primary factors limiting their growth in high nutrient low chlorophyll (HNLC) zones of the Southern Ocean
(Moore et al., 2013). Following a "middle of the road" SSP2-4.5 pathway, between 2015 and 2023 Southern Ocean
phytoplankton (defined as those south of 30°S per Gregg et al. (2003)) represented 36.31% of marine net primary productivity
globally, equivalent to 15.5 Pg C yr$^{-1}$ (Figure S1). Climate impacts on Southern Ocean phytoplankton are likely to manifest in
ecological shifts towards smaller cell sizes (Venables et al., 2013;Saba et al., 2014;Schofield et al., 2018;Biggs et al.,
2019;Mascioni et al., 2019) and changes in seasonal phenology (Moreau et al., 2015). Increases in overall productivity can be
most closely associated with a reduced duration and extent of sea ice coverage, allowing for a greater supply of irradiance to
surface waters of this light and iron co-limited productivity system. Strengthened upwelling is also likely to increase the flux
of existing iron supplies to the coastal (Annett et al., 2015) and open ocean (Moreau et al., 2023) from sedimentary or
hydrothermal sources, however, the extent to which changes in ocean mixing can be expected to impact nutrient supplies
remains largely unknown (Figure 1).

Shifts in community composition from diatoms to smaller cryptophytes have already been documented along the West
Antarctic Peninsula (Moline et al., 2004;Ducklow et al., 2007;Moline et al., 2008;Montes-Hugo et al., 2008;Rozema et al.,
2017), and are thought to be due to tolerance of cryptophytes to the low-salinity waters induced by increased sea ice melt
(Moline et al., 2004), or the tolerance of cryptophytes for high and variable light conditions in well stratified surface layers
(Mendes et al., 2023). Conversely, in culture-based competition experiments, diatoms are more successful in simulated future
ocean conditions over prevalent haptophytes such as *Phaeocystis antarctica,* albeit with reduced diatom cell sizes (Xu et al.,
2014). This difference is potentially driven by reduced iron limitation of diatoms and their greater tolerance to temperature
change (Zhu et al., 2016). These varied responses between manipulation experiments and *in situ* observations suggest that
physiological, as well as ecological, changes are important in understanding the net biogeochemical implications of
phytoplankton community change.

In the sea-ice zone, grazing by zooplankton accounts for ~90% of phytoplankton losses (Moreau et al., 2020). Shifts in
phytoplankton size class could rapidly cascade through the ecosystem as the dominant Southern Ocean zooplankton, Antarctic
krill (*Euphausia superba*, hereafter krill), are unable to graze small cryptophytes (Haberman et al., 2003), instead promoting
the dominance of carbon-poor salps (*Salpa thompsoni*), which reduces the overall efficiency of the marine food web (Ballerini
et al., 2014) and potentially weakens the biological carbon pump (Quéguiner, 2013;Biggs et al., 2021). Additionally, water
temperature, alongside changes to zooplankton abundance and diversity, has been shown to increase zooplankton metabolism
(López-Urrutia et al., 2006;Mayzaud and Pakhomov, 2014), which can in turn be expected to modulate the grazing pressure
and phytoplankton biomass (Lewandowska et al., 2014).



Projections of productivity in the Southern Ocean under future climate scenarios from the Coupled Model Intercomparison Project Phase 6 (CMIP6) class Earth System Models (ESMs) are actively informing research directions, International Panel on Climate Change (IPCC) reports (Masson-Delmotte et al., 2021), and governmental policy (Touzé-Peiffer et al., 2020). Yet, between CMIP5 and CMIP6 the spread of model projections with respect to vertical and horizontal physics as well as the number of phytoplankton functional types included has increased as different models incorporate more complexity and additional processes (e.g., varying elemental stoichiometry, phytoplankton diversity, complex elemental cycling) (Seferian et al., 2020). While representation of ocean physical drivers and nutrient fields compared to observations has improved between CMIP5 and CMIP6, surface chlorophyll is one of three key parameters that did not show improvement in benchmarking of CMIP6 performance over the global ocean (Canadell et al., 2021;Fu et al., 2022). Variance in model projections of phytoplankton and ocean biogeochemistry have been linked to the use of fixed C:N:P elemental stoichiometry (Kwiatkowski et al., 2018), an inability to reflect physiological adaptations, e.g. the ability of diatoms to maintain growth under iron limitation (Person et al., 2018), and complexities in modelling export fluxes, particularly in constraining phytoplankton losses through zooplankton grazing (Henson et al., 2022).

A major difference in the representation of productivity between CMIP6 models is the extent to which they consider different classes of phytoplankton. Diatoms (>20 µm) and pico-/nano-phytoplankton (predominantly cryptophytes and haptophytes) represent the vast majority of productivity across all latitudes of the Southern Ocean. Diatoms are a significant contributor to primary production and carbon export, accounting for ~40% of global marine primary production and POC exported to depth in the ocean (Jin et al., 2006;Tréguer et al., 2018). Diazotrophs (nitrogen-fixing phytoplankton) are present in small numbers, usually only in subtropical niches, due to the excess supply of nitrogen across the Southern Ocean (Luo et al., 2012). Calcifiers, mostly coccolithophores, inhabit waters north of 60°S where there is a strong supply of light but low Si, high Fe conditions, preventing the growth of diatoms (Charalampopoulou et al., 2016;Nissen et al., 2018). Only 11 CMIP6 models specifically include diatoms under future warming conditions and only three of these additionally consider pico-phytoplankton (CESM2, CESM2-WACCM and GFDL-ESM4).

In recent years, record low sea ice concentrations have been observed in the Southern Ocean (Raphael and Handcock, 2022;Turner et al., 2022). Given the dependence of Southern Ocean productivity on the timing of seasonal sea ice retreat, we consider it possible that this shift in trends of sea ice concentration could cause an abrupt change to sea ice dependent ecosystems in the near future (Swadling et al., 2023) . As phytoplankton are the main source of organic carbon in the Southern Ocean, uncertainty in projections of phytoplankton composition compounds existing model uncertainty in the biological carbon flux to the ocean's interior and seafloor (Henson et al., 2022). Within the context of climate change in the Southern Ocean, reducing model uncertainty in ecosystem mediated biogeochemical cycling will be of increased importance in determining the global scale impact of changes in the Southern Ocean productivity regime.



Here we aim to:

1.  Quantify the degree of uncertainty between models in projections of phytoplankton productivity with a SSP5-8.5 warming scenario, including different phytoplankton functional types.

2.  Determine mean trends between projected climate driven change in ecosystems, physical processes and biogeochemical cycling across different latitudinal zones of the Southern Ocean.

3.  Identify regions, timeframes and processes within the Southern Ocean Observing System (SOOS) framework (https://soos.aq), where the greatest projected changes and/or uncertainties occur. Here, we argue that targeting observations to establish phytoplankton-environment response interactions within the regions of the most rapid projected changes is essential to accelerate the improvement of phytoplankton representation in future generations of ESMs.

## 2. Methods

Model and observational data for the Southern Ocean were collected and visualised to determine a) the physical and biogeochemical changes that force or result from shifts in productivity, and b) the extent of primary productivity shifts over the next century in CMIP6.

### 2.1 CMIP projections

Model output was obtained from the Climate Model Intercomparison Project Phase 6 (CMIP6) data server via pangeo.io using the XMIP package in Python 3.11. Ensemble members for each parameter were chosen based on their availability for historical *(hist)* and SSP5-8.5 *(ssp585)* (ScenarioMIP) data (O'Neill et al., 2016). The selected models for each parameter are detailed in Table 1. Where an analysis type relied on the direct comparison between two or more parameters, only models that contained both parameters were selected. For the analysis presented in section 3.5, annual net primary production *(intpp)* is only included from models which also include diatom-specific annual net primary production *(intppdiat)* parameter. Where the same baseline model is included twice, because of having a low- and high-resolution version, the model is pre-averaged (i.e., both resolutions are assigned a weighting of 0.5 each) to avoid double counting of the same model when calculating the ensemble mean. Examples of models with two resolutions are highlighted in bold in Table 1. Only a small number of CMIP6 models contain irradiance limitation *(limirr)* and iron limitation *(limfe)* for multiple phytoplankton types, therefore analyses of light and iron limitation of phytoplankton utilise <5 models. All variables were extracted at monthly frequency, except for surface wind speeds where data were initially obtained daily; subsequently, annual weighted means were generated for most parameters per the weighting algorithm by Grover (2021). For mixed layer depth and incidental photosynthetically active radiation (IPAR), austral summertime means were used instead of annual means.



Model data were processed in Python 3.11 to apply the desired analysis (e.g., annual average, annual maximum) and then further averaged over residual variables (e.g., member_id). In most cases, all available member_id's were used; where this was not possible, any member_id's which could not be aggregated due to differences in array structure were removed. Net

primary production (NPP) is provided as a pre-integrated value across the water column; we integrated chlorophyll across the depth dimension between 0 and 500 m, to capture all phytoplankton across different depths, using the integrate function in SciPy (Virtanen et al., 2020). Subsequently, all models were re-gridded to a rectilinear grid via bilinear or nearest neighbour interpolation using XESMF (Zhuang et al., 2018) before being averaged to create multi-model means.

For spatial plotting, data were projected to the Antarctic Polar Stereographic (EPSG:3031) coordinate reference system in ArcGIS pro and visualised in QGIS using the Quantarctica package (Matsuoka et al., 2021), with post processing using SAGA and GDAL tools to remove imperfections in grid alignment through interpolation. All code to extract the CMIP6 data used in this study is available open access.



**Table 1: Selected models used in analysis of CMIP6 data. Models shown in bold represent multiple resolutions of the same core model.**

| Variable ID | Parameter | Units | Data selection | Models selected |
|---|---|---|---|---|
| *intpp* | Primary organic carbon production by all types of phytoplankton / diatoms | gC m$^{-2}$ yr$^{-1}$ | Annual average | ACCESS-ESM1-5, CanESM5, CanESM5-CanOE, CESM2 CESM2-WACCM, CMCC-ESM2, CNRM-ESM2-1, EC-Earth3-CC, GFDL-ESM4, GFDL-CM4, IPSL-CM6A-LR, MIROC-ES2L, **MPI-ESM1-2-HR**, **MPI-ESM1-2-LR**, MRI-ESM2-0, **NorESM2-LM, NorESM2-MM**, UKESM1-0-LL |
| *intppdiat* | | | | CanESM5-CanOE, CESM2-WACCM, CNRM-ESM2-1, GFDL-ESM4, IPSL-CM6A-LR , UKESM1-0-LL |
| *chl* | Mass concentration of total phytoplankton expressed as chlorophyll in sea water | kg m$^{-3}$ | Annual average | ACCESS-ESM1-5, CanESM5, CanESM5-CanOE, CESM2, CESM2-WACCM, CMCC-ESM2, GFDL-CM4, GFDL-ESM4, MIROC-ES2L, **MPI-ESM1-2-HR**, **MPI-ESM1-2-LR**, MRI-ESM2-0, **NorESM2-LM**, **NorESM2-MM,** UKESM1-0-LL |
| *limirrpico* | Irradiance limitation of pico-phytoplankton / miscellaneous phytoplankton / diatoms/ diazotrophs | Ratio of growth under environmental irradiance to growth under unlimited irradiance | Annual average | CESM2-WACCM, GFDL-ESM4 |
| *limirrmisc* | | | | CanESM5, CNRM-ESM2-1,GFDL-ESM4, IPSL-CM6A-LR |
| *limirrdiat* | | | | CESM2-WACCM, CNRM-ESM2-1, GFDL-ESM4, IPSL-CM6A-LR, UKESM1-0-LL |
| *limirrdiaz* | | | | CESM2-WACCM, GFDL-ESM4 |



| *limfediat/pic o/misc* | Iron limitation of diatoms/picop hytoplankton/ miscellaneous phytoplankton | Ratio of growth under environmental iron concentration to growth under unlimited iron concentration | Combined annual average | GFDL-ESM4 |
|---|---|---|---|---|
| *rsntds* | Net Downward Shortwave Radiation at Sea Water Surface (IPAR) | W m$^{-2}$ (Converted to µE m$^{-2}$ s$^{-1}$) | Summertime (daily) maximum | ACCESS-CM2, CanESM5, CanESM5-CanOE, CESM2-WACCM, CMCC-CM2-SR5, **CNRM-CM6-1**, **CNRM-CM6-1-HR**, CNRM-ESM2-1, EC-Earth3, EC-Earth3-CC, EC-Earth3-Veg, IPSL-CM6A-LR, MIROC-ES2L, **MPI-ESM1-2-HR**, **MPI-ESM1-2-LR**, **NorESM2-LM**, **NorESM2-MM** |
| *sfcWindmax* | Daily Maximum Near-Surface Wind Speed | m s$^{-1}$ | Annual average of daily maxima | AWI-CM-1-1-MR, BCC-CSM2-MR, CanESM5, CMCC-CM2-SR5, CMCC-ESM2, **CNRM-CM6-1**, **CNRM-CM6-1-HR**, CNRM-ESM2-1, EC-Earth3, EC-Earth3-CC, **EC-Earth3-Veg**, **EC-Earth3-Veg-LR**, GFDL-CM4, HadGEM3-GC31-MM, INM-CM4-8, INM-CM5-0, IPSL-CM6A-LR, KACE-1-0-G, **MPI-ESM1-2-HR**. **MPI-ESM1-2-LR**, MRI-ESM2-0, UKESM1-0-LL |
| *mlotst* | Ocean Mixed Layer Thickness Defined by $\sigma_t$ | m | Summertime maximum | ACCESS-CM2, BCC-CSM2-MR, CAMS-CSM1-0, CanESM5, CanESM5-CanOE, CESM2, CESM2-WACCM, CNRM-CM6-1, CNRM-ESM2-1, GFDL-ESM4, GISS-E2-1-G, HadGEM3-GC31-LL, IPSL-CM6A-LR, MPI-ESM1-2-HR, MRI-ESM2-0, NESM3, UKESM1-0-LL |
| *phos* | Sea surface pH | pH units | Annual average | CanESM5, CanESM5-CanOE, CESM2, CESM2-WACCM, GFDL-ESM4, IPSL-CM6A-LR, MIROC-ES2L, MRI-ESM2-0, NorESM2-LM |



| *tos* | Sea surface temperature | °C | Annual average | ACCESS-CM2, ACCESS-ESM1-5, BCC-CSM2-MR, CAMS-CSM1-0, CanESM5, CanESM5-CanOE, CESM2, CESM2-WACCM, CIESM, CMCC-CM2-SR5, CMCC-ESM2, **CNRM-CM6-1**, **CNRM-CM6-1-HR**, CNRM-ESM2-1, E3SM-1-1, EC-Earth3, EC-Earth3-CC, **EC-Earth3-Veg**, **EC-Earth3-Veg-LR**, FGOALS-f3-L, FGOALS-g3, FIO-ESM-2-0, GFDL-CM4, GFDL-ESM4, **HadGEM3-GC31-LL**, **HadGEM3-GC31-MM**, IITM-ESM, INM-CM4-8, INM-CM5-0, IPSL-CM6A-LR, KACE-1-0-G, KIOST-ESM, MCM-UA-1-0, MIROC6, MIROC-ES2L, **MPI-ESM1-2-HR**, **MPI-ESM1-2-LR**, MRI-ESM2-0, NESM3, **NorESM2-LM**, **NorESM2-MM**, TaiESM1, UKESM1-0-LL |
|---|---|---|---|---|
| *sios* | Surface concentration of silicic acid | $\mu$mol $L^{-1}$ | Annual average | CanESM5-CanOE, GFDL-ESM4, IPSL-CM6A-LR, **MPI-ESM1-2-HR, MPI-ESM1-2-LR, NorESM2-LM, NorESM2-MM**, UKESM1-0-LL |
| *no3os* | Surface concentration of nitrate | $\mu$mol $L^{-1}$ | Annual average | CESM2, CESM2-WACCM, GFDL-ESM4, NorESM2-LM, UKESM1-0-LL |
| *limno3* | Nitrate limitation of phytoplankton | Ratio of growth under environmental nitrate concentration to growth under unlimited nitrate concentration | Annual average | GFDL-ESM4 |



## 2.2 Regional data

Historical concentrations of surface nitrate and silicic acid plus sea surface temperatures were mapped from the World Ocean
Atlas 2018 data product (Garcia et al., 2019), representing average values from 1955 to 2017 . For Si*, annually averaged data
for nitrate and silicic acid were exported at a 1x1 degree resolution and subtracted from one another to produce Si*. To
determine Si*, pH and temperature values by SOOS area, SOOS regions south of 55° were drawn as mask layers and subset
using the zonal statistics function in QGIS.

## 3. Results and Discussion

**3.1 Physical climate drives biological changes in Southern Ocean water masses**

Climate change is driving substantial changes in Southern Ocean water masses (Bindoff et al., 2019). The widespread
strengthening of Southern Ocean winds by up to 0.8 m s$^{-1}$ (Figure 2a) and increased buoyancy fluxes (including freshwater
inputs) act as opposing drivers of stratification, modifying mixed layer depth (Figure 2b). Mixed layers are projected to deepen
across the Subantarctic by up to 10 m, but shoal across much of the rest of the Southern Ocean (Figure 2b). In light limited
regions a shoaling of the mixed layer can be expected to increase productivity, as phytoplankton become concentrated closer
to the surface, while in iron limited regions where iron is supplied by wintertime vertical mixing, deeper mixed layers can
benefit depth integrated primary productivity by increasing the productive water volume over which iron concentrations are
sufficient to promote growth (Llort et al., 2019) . Subsequently, the changing availability of light and iron across the Southern
Ocean determines the abundance and composition of primary producers. Despite the importance of changes in Southern Ocean
circulation for global ocean nutrient supply, the cumulative influence of physical processes across different spatial resolutions
results in poor overall performance of CMIP-class models in this region when their historical runs are compared with
observations (Meredith et al., 2019). A particular weakness of CMIP6 models is in reconstructing the sea ice changes that
drive buoyancy forcing (Roach et al., 2020;Shu et al., 2020) which has an important role in determining the flux of heat and
CO$_2$ across the ocean-atmosphere boundary. The uncertainty in sea ice change also results in a high degree of variation in
coastal irradiance between models (Figure S2e), particularly for the Weddell Sea and Ross Sea regions. Recent large and
unexpected changes in sea ice around Antarctica emphasise that greater knowledge of the key drivers and controls is required,
in order to improve predictive skill in models (Turner and Comiso, 2017).

Across the Southern Ocean, the timing of the springtime onset of net primary production and the magnitude of summer biomass
accumulation are controlled by light availability, as dictated by sea ice extent, cloud cover and water column structure (Henley
et al., 2017). CMIP6 models project the greatest increase in productivity to occur across the coastal zone of the Southern Ocean
(65-90°S) (Figure 2c), where irradiance limitation is reduced (Figure 2d). Conversely, across the Transitional zone (40-50°S),



IPAR reduces (Figure 2e), irradiance limitation increases (Figure 2d) and productivity increases are less here compared to the rest of the Southern Ocean (Figure 2c). Increased iron limitation (Figure 2f) likely manifests from greater competition for iron driven by increased productivity (Figure 2c) despite a potential increase in iron supply with a deepening of the mixed layer across parts of the Subantarctic (50-65°S) (Figure 2b); brought about by reduced upper-ocean stratification from strengthening zonal winds (Carranza and Gille, 2015;Sallee et al., 2021). Iron supply to the surface is subject to changes in the properties and movement of water masses, which lead to variable circulation strengths, depth boundaries, heat content and carbon sequestration resulting from climate-driven perturbations to the ice-ocean-atmosphere system (Bindoff et al., 2019;Meredith et al., 2019). Upwelling of nutrients and light availability for phytoplankton are both strongly influenced by mixed layer depth, which in turn varies seasonally with increased solar warming and ice melt driving deeper Southern Ocean pycnocline stratification through the summer (Sallee et al., 2021). Models generally agree on changes in summertime mixed layer depth across most of the open ocean (Figure S2b), the greatest source of uncertainty is at the terminus of the Ross and Flicher-Ronne ice shelves, inclusion of freshwater input from ice shelves in future CMIP generations could help to reduce uncertainties in stratification.





**Figure 2: CMIP6 anomaly representing change to the end of the century in A) near-surface wind speed, B) mixed layer depth, C) net primary productivity, D) irradiance limitation of phytoplankton, E) incidental photosynthetically active radiation (IPAR) , and F) iron limitation of phytoplankton. Changes are calculated from an ensemble of CMIP6 models, comparing a historical (1985-2015) average against 2090-2100 under the SSP5-8.5 climate scenario. Details of ensemble members are given in Table 1. *Units in panels D and F are arbitrary ratios of growth under environmental irradiance or iron concentrations against potential growth under unlimited irradiance or iron concentrations. Positive values represent an increase in limitation, while negative values represent a decrease in limitation.**




### 3.2 Changing biogeochemistry of the Southern Ocean

**3.2.1 Micronutrient supply and uptake**

Iron acts as the primary limiting nutrient across the Southern Ocean (de Baar et al., 1995;Watson et al., 2000), due to supply limitation from low atmospheric inputs and significant distances from terrigenous sources (Boyd and Ellwood, 2010). Around the Antarctic coast, iron concentrations are set by processes including the resuspension of shelf sediments (Blain et al., 2001), melting of sea ice (Lannuzel et al., 2016) and potential transformation of iron into more labile forms by glacial retreat, as seen in the Arctic (Laufer-Meiser et al., 2021). The change in projected iron limitation of phytoplankton appears minimal (between -12 and 10% Figure 2f). Iron limitation is expected to increase most in the transitional zone between South America and New Zealand, correlating with a reduction in near-surface wind speed (Figure 2a), suggesting that atmospheric deposition of iron will decline in this region. There is a minor increase in iron limitation around the Antarctic coast, which represents the inverse of the decreasing trend in irradiance limitation (Figure 2d), indicating a shift towards an increasingly iron limited system, as reductions in sea ice concentrations increase light availability to coastal waters. The co-occurrence of an increase in iron limitation (Figure 2f) and an increase in total productivity (Figure 2c) across the coastal zone suggests that this increase in limitation is driven by an increase in the uptake of iron (from a larger productivity sink), as opposed to any substantial changes in supply.

Despite the importance of iron for phytoplankton growth in the Southern Ocean, CMIP series models have historically struggled to resolve the vertical supply of  dissolved iron (Tagliabue et al., 2016), which results in widespread uncertainty for modelling primary and export productivity. At the group level, iron limitation could be expected to influence shifts in phytoplankton communities because larger cells have a greater demand for iron compared to smaller cells. In addition, other micronutrients such as manganese have been identified as a control on phytoplankton growth, particularly during seasonal transitions (Browning et al., 2021). For example, manganese has been shown to play an important role in controlling oxidative stress by catalysing antioxidant production in some diatom species (McCain et al., 2021), explaining the observed phenomenon of iron-manganese co-limitation in the Southern Ocean (Pausch et al., 2019;Browning et al., 2021;Balaguer et al., 2022). Yet only iron is considered in ESMs, due at least partially to the lack of observational data to underpin distribution modelling of other micronutrients. Future work should continue to develop our understanding of the metabolic role of other micronutrients and additionally consider the extent to which diversity exists in micronutrient demand among Southern Ocean phytoplankton species.

**3.2.2 Macronutrient supply and uptake**

Nitrogen species, silicic acid (DSi) and phosphate are all essential for the growth and survival of diatoms, with nitrate and phosphate also being required by all other phytoplankton classes for cellular metabolism. The ratio of utilisation between



nitrogen (N) and phosphorus (P) deviates from the Redfield (1958) ratio of 16:1 across the Southern Ocean according to changes in community composition (Weber and Deutsch, 2010;Henley et al., 2020). Unlike much of the global ocean (Moore et al., 2013), high rates of macronutrient supply from the Circumpolar Deep Water (CDW) prevent widespread N or P

limitation in the Southern Ocean except in periods of intense summer growth in high-productivity coastal regions (Henley et al., 2017). Although projections indicate an increase in chlorophyll across such regions (Figure 3), models do not show any increases in nitrate limitation over the remainder of the century (Figure S3), suggesting that iron and light will continue to be the primary constraints on productivity.

While macronutrients are not usually limiting to Southern Ocean phytoplankton, growth of diatom communities, particularly around high productivity coastal and island zones (supported by lateral iron advection) (Robinson et al., 2016), is likely to place an increased demand on DSi availability (Table 2). The relationship between Si and N is denoted as Si* ($[Si(OH)_4]$–$[NO_3^-]$) (Sarmiento et al., 2004), with high Si* values (> 25) indicating plentiful DSi availability that supports diatom growth, while low values (< 10) suggest conditions which favour non-silicifying phytoplankton, such as the smaller cryptophytes and

haptophytes. Si* is highest in the Antarctic zone (Henley et al., 2020) because of silica input from upwelling of CDW, but remains spatially heterogeneous within this region (Table 2). Si* is consistently high in the Weddell Sea, while across the WAP and Ross, Amundsen and Bellingshausen Seas there is a moderate mean Si* with large variability, and the Indian Sector has a substantially lower DSi availability. Si* is projected to decline by 2090-2100 at a zonally averaged value of -8.5 μmol $L^{-1}$, with the greatest declines being in the Ross and Weddell Seas, as well as the Indian sector (Table 2).


Changes in Si* correlate with increases in chlorophyll concentration across the same regions (Figure 3), indicative of increased phytoplankton concentrations resulting in a drawdown of silicic acid. However, increases in chlorophyll appear independent from projected changes in primary productivity (Figure 2c). For example, the west Antarctic Peninsula and Amundsen Sea regions show the greatest increase in primary productivity, but are among the regions of smallest change for both Si* and

chlorophyll. The divergence between chlorophyll and primary productivity indicates variability in Chl:C, with siliceous diatoms typically expressing more chlorophyll per unit of carbon (Sathyendranath et al., 2009). Therefore, the large chlorophyll increase and large Si* decline projected in the Weddell Sea is likely driven by an increase in diatoms, whereas the productivity increase with only small changes in both chlorophyll and Si* seen on the west Antarctic Peninsula probably results from an expansion of non-diatom phytoplankton with lower Chl:C.


The impact of climate change on DSi supply to the surface is difficult to evaluate because it is dependent on the competing stratification effects from wind driven changes to upwelling and an increase in freshening. Export of DSi from the surface and remineralisation at depth additionally act as important controls on supply; Freeman et al. (2018) showed that increased biological uptake of silicic acid, through increased diatom growth, leads to a poleward shift in the silicic acid front and a

potential decoupling from the Antarctic polar front. Efforts to better define the nutrient budgets, particularly in increasingly



common low sea ice years, across different sectors of the Southern Ocean, as well as understanding the changing nutrient demands of phytoplankton will be essential for determining future trends in nutrient limitation (Henley et al., 2019).

### 3.2.3 Ocean acidification

Across all regions of the Southern Ocean, continued uptake of anthropogenic $CO_2$ is expected to elicit a decrease in surface
pH of ~0.45 units south of 55°S (Table 2) under the high emission scenario (SSP5-8.5). Projected changes in pH do not differ regionally, and shows little variation within regions (low standard deviation). This ubiquity in the acidification effect means there no evidence for a direct effect of acidification effect on phytoplankton within our assessment of the Southern Ocean. In the main biogeochemistry modules of CMIP6 members, phytoplankton growth is driven and limited by nutrients and light, while many models are built with complex carbonate systems, these typically only interact with rates of calcification and for
the majority of phytoplankton there is no biotic feedback from changes to the carbonate system. Although ocean acidification (OA) is typically considered to have the greatest effect on marine calcifiers through impacts on the production and dissolution of calcium carbonate (Figuerola et al., 2021), OA is also likely to impact diatom (Petrou et al., 2019), pico-phytoplankton (Tortell et al., 2008) and krill (Kawaguchi et al., 2013) populations which form the base of Southern Ocean food webs.

When considering species shifts, unravelling the specific impact of OA on individual phytoplankton types is complex due to the fact that OA often acts on phytoplankton indirectly. For example, Petrou et al. (2019) showed that acidification reduces silicification of diatoms (Si:C), likely reducing sinking capacity and increasing rates of remineralisation in the upper ocean, in turn weakening ocean carbon drawdown and acting as a positive feedback on the carbon cycle. Yet in existing CMIP6 biogeochemical modules, silicification is a product of the ambient concentration of either silicic acid (Stock et al., 2020), or
silicic acid and iron (Moore et al., 2004), with no interaction from OA. This lack of plasticity might also impact nutrient export rates, Taucher et al., (2022) showed that OA decreases Si dissolution in the surface ocean, resulting in an export flux with a higher Si:N composition. Expanding model setups in CMIP6 to include the impact of OA and other physiochemical drivers on biogeochemical stoichiometry could help to resolve existing biases in Southern Ocean silicic acid concentrations (Long et al., 2021), which has the potential to cause large uncertainty in global NCP due to the role of the Southern Ocean as a net Si
source to the global thermohaline circulation. A key challenge in developing our understanding of OA impacts on phytoplankton at the group level is in resolving the mechanistic influence changing acidity has on different phytoplankton. For example, productivity may increase under OA as a result of shifts towards larger diatom species (Tortell et al., 2008). However, some studies have shown that smaller size classes (<20 μm) are more successful at higher $pCO_2$ values (Hancock et al., 2018) or that total productivity may decrease under OA (Westwood et al., 2018).




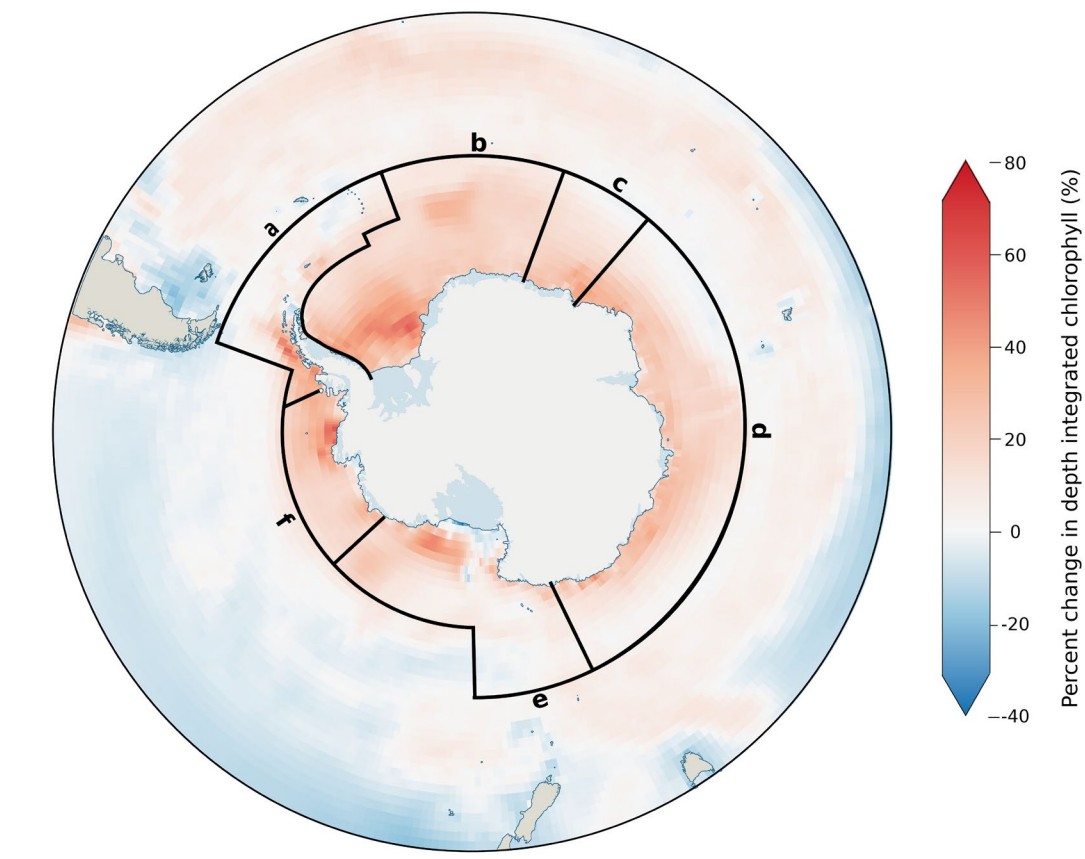

**Figure 3: Change in depth-integrated chlorophyll (0-500 m) from all phytoplankton, displayed as the percentage change between the annual historical average (1985-2015) and projected values for 2090-2100. Values shown are multi-model means of the models listed in Table 1. Spatial boundaries show the Southern Ocean Observing System (SOOS) regions south of 55°S, which are defined in Table 2.**





**Table 2: Biogeochemical parameter values calculated for the Southern Ocean Observing System regions. SOOS regional working groups (as defined at: www.soos.aq/activities/rwg) indicated on Figure 3; section C is an overlap section of sections B and D. Data shown are: Si\* ([Si(OH)₄]−[NO₃⁻]) values and temperature determined from objectively analysed annual means of World Ocean Atlas 2018 data. pH was determined from a historical run of a multi-model ensemble of CMIP6 models (1985-2015). Delta values are anomalies of multi-model means of pH, temperature and Si\* based on comparisons between the mean annual historical value (1985-2015) and projected values for 2090-2100 under SSP5-8.5 for a CMIP6 ensemble (detailed in Table 1). Values in brackets are standard deviations, representing spatial variation across the region. Anomaly maps for pH, temperature and Si\* are shown in Figure S4, S5 and S6 respectively.**

| Section | SOOS Region | Si* (µmol/L) | ΔSi*(µmol/L) | pH | Δ pH | Temperature (°C) | Δ Temperature (°C) |
|---------|-------------|--------------|--------------|-----|------|------------------|--------------------|
| A | West Antarctic Peninsula & Scotia Arc | **17.24** **(17.82)** | **-7.06** **(3.63)** | **8.07** **(0.15)** | **-0.43** **(0.01)** | **1.94** **(1.87)** | **1.79** **(0.37)** |
| B | Weddell Sea & Dronning Maud Land (WSDML) | **37.37** **(9.70)** | **-10.98** **(3.77)** | **8.08** **(0.14)** | **-0.41** **(0.01)** | **-0.07** **(0.95)** | **1.43** **(0.49)** |
| C | SOIS/WSDML | **23.16** **(6.67)** | **-8.20** **(1.59)** | **8.07** **(0.01)** | **-0.41** **(0.01)** | **1.08** **(1.26)** | **1.89** **(0.40)** |
| D | Southern Ocean Indian Sector (SOIS) | **4.71** **(3.72)** | **-10.52** **(3.67)** | **8.07** **(0.13)** | **-0.41** **(0.01)** | **1.78** **(1.66)** | **1.78** **(0.46)** |
| E | Ross Sea | **19.82** **(18.49)** | **-9.13** **(5.02)** | **8.08** **(0.13)** | **-0.40** **(0.02)** | **0.62** **(2.22)** | **1.08** **(0.46)** |
| F | Amundsen and Bellingshausen Seas | **17.59** **(14.02)** | **-5.27** **(4.33)** | **8.09** **(0.16)** | **-0.43** **(0.01)** | **0.11** **(0.83)** | **1.73** **(0.44)** |

## 3.3 Primary production and representation in CMIP6

### 3.3.1 Phytoplankton classes

Two CMIP6 models (GFDL-ESM4 and CESM2-WACCM) showed substantial mechanistic differences in productivity projections by phytoplankton type south of 65°S (Figure 4). While GFDL-ESM4 projects that in this region diatoms account for the majority (55%) of the change in productivity under SSP5-8.5 (Figure 4a,b), diatoms represent only 26% of the productivity increase in CESM2-WACCM, with pico-phytoplankton forming the major (74%) phytoplankton group (Figure





4e,d). Additionally, the GFDL model indicates that increased productivity is driven by increases in both diatoms and pico-phytoplankton, representing a simultaneous growth scenario while CESM2-WACCM favours a replacement mechanism with diatoms decreasing as pico-phytoplankton populations grow (Figure 4c,f). In CESM2-WACCM (MARBL biogeochemistry module) and GFDL-ESM4 (COBALTv2 biogeochemistry module), growth of phytoplankton groups is a product of temperature, nutrient limitation and light availability. In COBALTv2 the iron uptake half saturation constant is greater than in MARBL (0.1 vs 0.03 nmol kg$^{-1}$ for small phytoplankton) and the differential between small and large phytoplankton (diatom) iron requirements is greater (x5 vs x2.3). Although phytoplankton in MARBL have lower Fe requirements, negative biases towards $NO_3^-$ and $PO_4^{-3}$ in the Southern Ocean by CESM2 suggest that NCP is overestimated, subsequently this could drive the system to iron limitation earlier, resulting in an "insufficient contribution from diatoms" (Long et al., 2021). This could suggest that GFDL-ESM4 presents a more realistic outlook for phytoplankton composition, however fixed nutrient constants which usually represent a global average collected from multiple studies, make no differentiation for changes to nutrient uptake in cold water environments. For example, Timmermans et al. (2004) showed iron uptake half saturation values for Southern Ocean diatoms to vary substantially between 0.19 and 1.14 nmol L$^{-1}$ based on species, compared to a fixed value of 0.5 nmol L$^{-1}$ for COBALTv2 and 0.07 for MARBL, representing diatoms globally (Stock et al., 2020;Long et al., 2021). Experimentally, uptake half saturation constants are determined through the sequential addition of nutrients, yet multiple studies have shown that Southern Ocean diatoms in particular are able to reduce their cellular iron demand through changes to the photosynthetic pathway (e.g. Strzepek and Harrison, 2004;Jabre et al., 2021). Therefore, models based on these fixed constants may be reflecting the maximal iron uptake rather than the low iron acclimated uptake, i.e., this approach towards modelling nutrient limitation does not allow for a molecular adaptation which can, in some cases, achieve the same growth rate under more limiting conditions.

Key to understanding the ecological and biogeochemical impact of molecular level adaptations will be a shift from single species to community based experiments alongside incorporation of holistic marine ecosystem models which account for an expanded range of biological interactions involved in phytoplankton-zooplankton predation and bacterially-driven mixotrophic effects, which can substantially alter trophic energy transfer and export fluxes (Ward and Follows, 2016). Trait-based approaches have been explored as a means of modelling phytoplankton community composition, distinguishing functional groups based on life histories, morphology and physiology (Litchman and Klausmeier, 2008). Ocean biological sampling has some of the lowest coverage in the Southern Ocean (Sunagawa et al., 2020). Expansion of ecosystem observing at the metagenomics level (e.g., Guidi et al., 2016) offers a promising opportunity to expand our knowledge of traits and trade-offs in Southern Ocean phytoplankton communities, facilitating their integration into climate models.



### 3.3.2 Ecological dynamics and ecophysiology

In a changing ocean, phytoplankton will succeed where they have the greatest biological plasticity, for example the ability to
photo-acclimate rapidly (Arrigo et al., 2010) or scavenge and utilise a diverse range of micronutrients. The physiological
properties of any individual species ultimately determines their ability to survive in a particular region at a particular time
under ever changing climate-driven conditions. Subsequently, species ecology determines the abundance and temporal extent
with which a species can exist or compete in a particular region. As the warming of the climate continues to bring about an
earlier retreat of sea ice, growth seasons are expected to lengthen, altering the temporal dynamics of species progression
(Moreau et al., 2015). In the coastal zone of the Southern Ocean, changes in light appear to be the main influence on
productivity with decreased irradiance limitation stimulating pico-phytoplankton growth to a greater extent than diatoms
(Figure 4 j,k,m,n); meanwhile iron limitation shows little correlation with productivity changes in this region (Figure 4 g,h),
likely because of replete iron supplies from coastal upwelling.





**Figure 4: Evaluation of GFDL-ESM4 and CESM2-WACCM models using an anomaly between 2090-2100 (SSP5-8.5) and a historical average (1985-2015) for the Antarctic zone (65-90°S). Linear regression between change in total productivity and pico-phytoplankton productivity for GFDL-ESM4 (A) and CESM2-WACCM (D). Linear regression between change in total productivity and diatom productivity for GFDL-ESM4 (B) and CESM2-WACCM (E). Linear regression between change in picophytoplankton productivity and diatom productivity for GFDL-ESM4 (C) and CESM2-WACCM (F). Change in iron limitation with picophytoplankton (G) and diatom (H) productivity for GFDL-ESM4. Change in irradiance limitation with pico-phytoplankton (J) and diatom (K) productivity for GFDL-ESM4. Change in irradiance limitation with pico-phytoplankton (M) and diatom (N) productivity for CESM2-WACCM.**





### 3.4 Latitudinal productivity projections in CMIP6.

From those models that do distinguish between at least diatoms and other phytoplankton we are able to examine

projected changes in community composition over the 21$^{st}$ century under a continued warming scenario (SSP5-8.5) (Figure 5). Previous analysis by Laufkötter et al. (2015), using a different set of models (a mix of marine ecosystem models employed in CMIP5 and the Marine Ecosystem Model Intercomparison Project), found substantial disagreement between models in projecting which phytoplankton groups drove NPP changes in the Southern Ocean. In CMIP5, Leung et al. (2015) found a latitudinally banded response of phytoplankton to

continued warming, driven by the bottom-up dynamics of nitrate, iron and light limitation. From this analysis, we applied the same latitudinal bands to our analysis of the changes in whole community, diatom and non-diatom productivity across CMIP6. Our whole community projections agree with the trends shown by Leung et al. (2015), of a poleward increase in phytoplankton productivity, increasing average total productivity south of 40°S, with increases in total productivity in the Transitional (40-50°S; Figure 5b), Subantarctic (50-60°S; Figure 5c), and

Antarctic zones (65-90°S; Figure 5d). In relative terms, this reflects a ~10% increase in total productivity over the SSP5-8.5 run (2015-2100) for both the Transitional and Subantarctic zones, with a ~30% increase in productivity for the Antarctic zone (Figure S7 b,c,d). The poleward increases in productivity correlate with a deepening of mixed layers around the Antarctic zone (Figure 2b) and a reduction in coastal light limitation (Figure 2d), resulting in greater increases in Antarctic zone productivity compared to the Transitional or Subantarctic. An ensemble

mean shows no overall change in productivity across the Subtropics (see also Tagliabue et al., 2021), however individual models show the widest degree of divergence in this region, with some models projecting decreases in the diatom population of over 60% (Figure S7a), indicating a large amount of uncertainty in the magnitude of productivity changes.

In the Subantarctic (50-65°S), despite a large projected increase in light availability (Figure 2e), models project only a minor increase in productivity driven by a small amount of diatom growth, suggesting that growth of both diatom and non-diatom species remains largely iron-limited in this region. The coastal zone shows the greatest degree of change in phytoplankton growth, with the largest increases in this region; the majority of the biomass change can be attributed to non-diatoms, however relative changes in both diatom and non-diatom populations are similar (Figure S7d), suggesting no overall changes to community composition here. The continued increase

in all phytoplankton classes can be attributed to the decreased iron limitation across much of the zone (Figure 2f), with the success of non-diatoms reflecting the increase in light limitation (Figure 2d). Large phytoplankton types



are more strongly affected by light limitation in CMIP6 because they have a greater requirement for light (as a lower constant for the chlorophyl specific initial slope of the photosynthesis-irradiance curve), in COBALTv2,

large phytoplankton require 3x as much light as small phytoplankton to reach the same rate of photosynthesis (Stock et al., 2020). The lower requirements for iron and light by smaller phytoplankton types means that the change in relative abundance of smaller phytoplankton types to environmental change is often greater compared to diatom populations.

While CMIP6 models do not explicitly consider phytoplankton size, the shift from diatoms to, typically smaller, non-diatom species is consistent with more advanced ecological models such as DARWIN which predict a decrease in the slope of the phytoplankton size spectrum, albeit over a greater area of the Southern Ocean than shown in CMIP6 (Henson et al., 2021). Despite the clear differences between latitudinal bands, spatial heterogeneity continues to exist within these zones, particularly for the coastal zone where some of the greatest

increases in chlorophyll occur in the WAP and Weddell Sea regions (Figure 3), reflecting the disproportionately high DSi supply in these regions (Table 2). Resolving spatial heterogeneity of phytoplankton in global-scale models such as those in CMIP6 is likely to require an increased reliance on, and integration with, regional-scale modelling (Person et al., 2018). The rapid increase of non-diatom species around the coast is in agreement with studies describing declining large diatom (>20 μm) abundances (Kang et al., 2001;Wright et al., 2010;Pearce et

al., 2011); however, while it is true that diatoms are projected to decrease as a proportion of the community, diatom-derived carbon production is still projected to increase under continued warming, suggesting that the coastal biological carbon pump may be less threatened by this shift in community composition than previously thought.





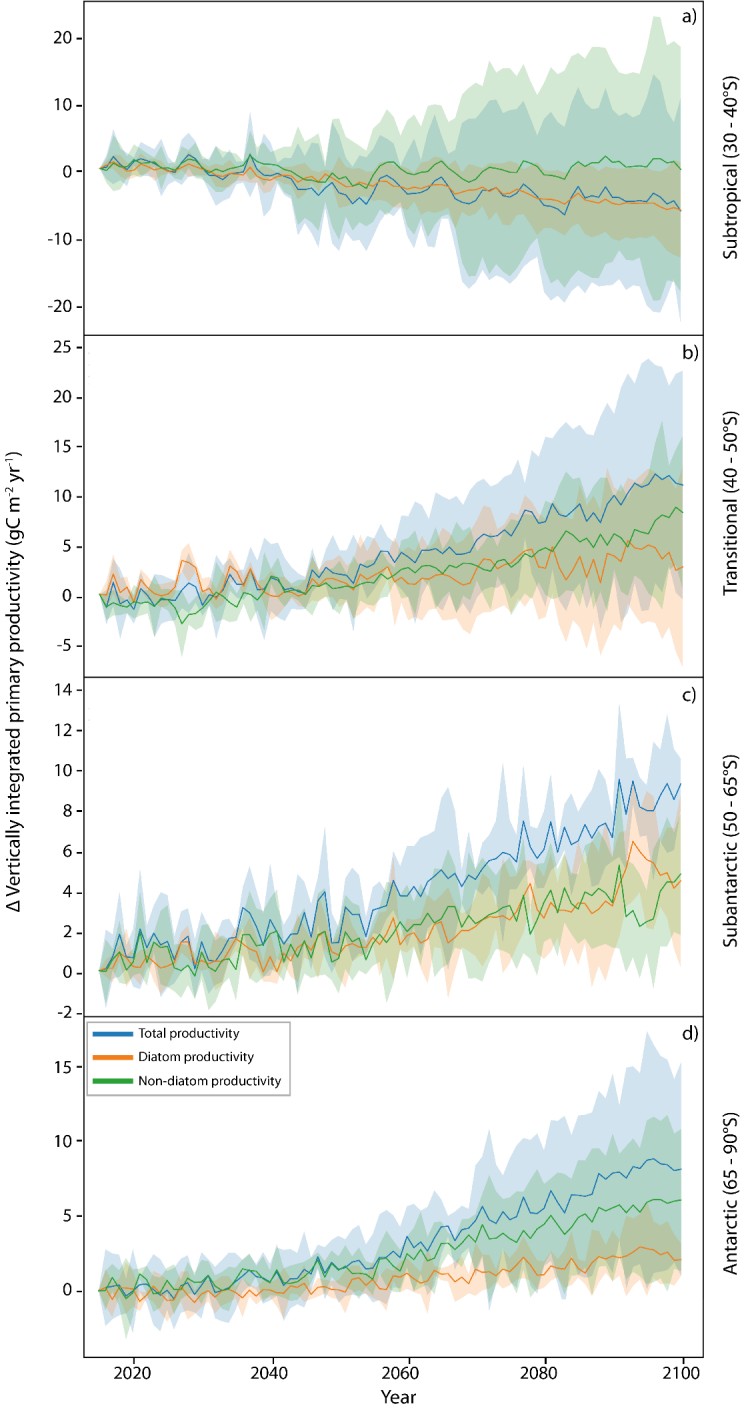

Figure 5: Changes in productivity (g C m⁻² yr⁻¹) and the contribution of different phytoplankton classes to productivity, 2015-2100. The anomaly in CMIP6 model productivity projections (as POC production) compared to 2015 for SSP5-8.5 conditions across 4 latitudinal bands of the Southern Ocean, per Leung et al. (2015). Lines represent multi-model means of total productivity (intpp), diatom productivity (intppdiat) and non-diatom productivity (intpp-intppdiat). Shaded regions represent the spread between models as the interquartile range. Six CMIP6 models were used in this analysis, because only models containing the diatom productivity parameter are included; details of the specific models assessed are given in Table 1.



## 4. Conclusions

### 4.1 Implications of Southern Ocean productivity shifts

The cumulative impact of climate change on phytoplankton has the potential to restructure ecosystems of the Southern Ocean, with wider consequences for global ocean productivity and climate. In this study we found that CMIP6 models project a future Southern Ocean with increased levels of productivity, particularly around the Antarctic coastal zone. The major driver of this is reductions in light limitation, brought about by increased light concentrations from reduced sea ice coverage. However, the extent to which light will change is a source of great uncertainty, with the poor performance of sea ice also having additional implications for buoyancy forcing. Resolving freshwater fluxes from the AIS could reduce model uncertainty in coastal mixed layer depth change. The current iteration of CMIP6 models does not suggest any significant shifts in community composition across the Southern Ocean, outside of a decrease in the relative abundance of diatoms in the subtropics. However, there is a large uncertainty of up to ± 30% for all phytoplankton classes across all zones of the Southern Ocean, and key processes which will impact phytoplankton (e.g., viral losses, composition of the grazer community) are absent from most models.

In models which do separate productivity by phytoplankton type, the growth of one type of phytoplankton over another is a product of light, temperature and nutrient limitation. Fixed nutrient stoichiometry is a key limitation in projecting phytoplankton composition, in particular the lack of interaction between physiochemical processes such as OA with biogeochemistry could be responsible for some of the existing biases in models towards excess Si in the Southern Ocean. We showed that for two models (GFDL-ESM4 and CESM2-WACCM) the iron requirements of different phytoplankton types can result in either a simultaneous growth of diatoms and picophytoplankton, or a replacement of diatoms with picophytoplankton. Future model generations might consider the acclimation of diatoms to low iron conditions, models currently use very different uptake half saturation values for iron, which disproportionally impacts community composition in iron limited regions. However, the literature also contains wide divergence in experimentally determined nutrient uptake constants, a more accurate approach towards modelling will first require us to better define what drives this variability in iron requirements. With continued record low sea ice trends, observation of phytoplankton responses in this multi-stressor environment will be essential in understanding the scale of productivity change occurring and provide a basis to incorporate phytoplankton community change into global scale modelling efforts.

### 4.2 Observational recommendations:

We have identified changes in nutrient upwelling (both upwelling strength and concentration of nutrients at the surface), directionality of mixed layer depth change, mutualism and resource competition between phytoplankton classes, and eddy strengthening as key processes that require improved representation in future generations of climate models. In some cases

this is because of lack of monitoring and insufficient data coverage to assess temporal (e.g. seasonal nutrient dynamics) or spatial (e.g. biogeochemical impacts of eddies) variability. While determining the magnitude of MLD change that can be associated with freshwater injection, variability resulting from the relatively recent reversal in Antarctic sea ice trends poses a particular challenge to resolving this process.


Some of the greatest uncertainties in phytoplankton composition exist in diatom abundance between 30°S and 65°S; observationally this may be related to data sparsity, with a greater density of measurements resulting from land-based stations in the coastal zone which has a lower relative uncertainty. However, the existence of a single diatom group in models that consider different phytoplankton functional types could also be limiting here, as subtropical diatoms differ in biology (e.g., lack of ice binding proteins) and species composition compared to polar diatoms. Parameterising these distinct species groups in one functional type inevitably means the model will underperform for some groups relative to others. Monitoring of phytoplankton functional group composition, especially in areas of the greatest projected change (WAP and Weddell Sea), ideally capturing temporally-resolved new productivity in the early spring, will be necessary to detect any changes in community composition. The relationship between chlorophyll, carbon production and Si* appears to be an indicator of community composition, with projected increases in chlorophyll and decreases in Si* in the Weddell Sea indicating a diatom driven increase in productivity. Proxy variables such as these may allow for a coarse determination of community composition from autonomous platforms without the need for personnel intensive microscopy or genomic methods. Crucially, to then link ecological shifts to global climate, the extent to which different phytoplankton species undergo losses via different pathways represents a substantial knowledge gap that needs to be filled to improve the predictive skill of ESMs.




Finally, there is a need for process-based studies including a diverse set of phytoplankton species to further our understanding of fundamental life cycle processes and ecological-biogeochemical coupling. A focus towards defining variability in phytoplankton buoyancy, carbon uptake and release as dissolved organic matter, grazing by zooplankton, interactions with viruses, macronutrient stoichiometry and micronutrient utilisation will enhance our core understanding of Southern Ocean phytoplankton dynamics in a changing climate.


**Acknowledgements**

BJF is supported by a NERC Doctoral Training Partnership Grant (NE/S007407/1). SFH would like to acknowledge support from NERC (NE/K010034/1). This work used JASMIN, the UK's collaborative data analysis environment (https://jasmin.ac.uk). For the purpose of open access, the author has applied a creative commons attribution (CC BY) licence to any author accepted manuscript version arising.




**Data availability**

CMIP6 data were obtained through, and are freely available at panego.io. Specific models and parameters extracted for each
analysis are listed in Table 1. Code for CMIP6 data analysis are available at: *(zenodo link to be later inserted, available to reviewers as attached zip file)*.

**Author contributions**

BJF, SFH, MPM and AJP devised the concept for the paper and contributed towards initial drafting and editing. BJF performed
model analysis and produced the figures. OS and KB provided input on conceptual design and edited previous versions of the
manuscript. All authors have approved the final version of the manuscript.

**Competing interests**

The authors declare no competing interests.

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
