# Peer review of "Climate driven shifts in Southern Ocean primary producers and biogeochemistry in CMIP6 models"

_EGUsphere, 2024_

## Author Response (AR1)

**Author Response to Reviewer 1:**

**Reviewer comments in black**
**Author responses in blue**

This study focuses on climate-driven shifts in Southern Ocean phytoplankton composition and primary production. It provides an analysis of Southern Ocean phytoplankton communities and biogeochemistry simulated in CMIP6 models, making it a contribution to the growing literature on biogeochemistry and future trends in the Southern Ocean.

First of all, I appreciate the massive work done by authors. Although I found the analysis valuable, I have four major concerns about the manuscript that suggest it could benefit from some restructuring. Here are the main areas I believe need attention.

We thank the reviewer for their comments and appreciation of the contribution of our work. In the responses below we set out our approach to addressing their concerns.

1. **Connection Between Text and Figures/Tables:** I sometimes had difficulty following the connections between the text and the figures/tables. Therefore, I raised some questions and made some suggestions in the comments below.

   We thank the reviewer for their suggestions to improve clarity in connecting the text with figures and tables, we have made changes in response to the specific points raised in the comments below.

2. **Results and Discussions:** While the discussions on mixed layer depth, light availability, and iron limitation are there, the manuscript would benefit from incorporating additional points. Specifically, more discussion is needed on the roles of grazers and changes in the Antarctic ice sheet and their effects on micro and macronutrients. It is well-known that changes in the Antarctic ice sheet will impact nutrients such as iron and silicate (e.g., Wadham et al., 2019; Death et al., 2014; Tréguer, 2014). However, this point is not mentioned in the paper. Is there a specific reason for that related to ESMs?

   We agree with the reviewer about the biogeochemical importance of the Antarctic Ice Sheet (AIS). The reviewer is correct that ESM setup is the main reason that we do not mention changes in nutrient fluxes from the AIS, simply the AIS is not represented in CMIP6 models so there can be no direct changes to either the biology or the biogeochemistry which result from climate driven changes in the AIS. *Purich and England* [2021] summarise this as "the majority of models have a 1° horizontal resolution and as such do not resolve the Antarctic Slope Current (ASC), eddies and tides are not generally resolved, ice shelves are not resolved and meltwater from ice shelves and sheets is not included". We have added an explanation of this to the manuscript.

   Lines 214-218: "This uncertainty in stratification can be linked to the lack of representation of ice shelves and their meltwater flux in the current generation of CMIP models [*Purich and England*, 2021]. Subsequently, the lack of a meltwater flux directly impacts biology and biogeochemical cycles through the absence of ice associated nutrient seeding [e.g. *Death et al.*, 2014] alongside creating uncertainty in nutrient and light availability through an incomplete representation of stratification."

   Additionally, while light and iron limitations are crucial in the Southern Ocean, the impact of grazers on phytoplankton composition is also significant (e.g., Smatecek, 2004). Grazers control phytoplankton bloom phenology in general (e.g., Banse 1994; Banse, 2013), and extensive analyses on zooplankton in CMIP6 simulations exist (e.g., Rohr et al., 2023; Petrik

et al., 2022). Including more discussion on this aspect would benefit the manuscript. The summary of climate change pressures and their biogeochemical consequences for ecosystem productivity in Figure 1 is good. However, parts that do not have numbers need better integration into the manuscript's results and discussion sections. Revising this for a more concise format would strengthen the manuscript.

Grazers are definitely a very important part of the ecosystem and we refer to zooplankton communities throughout. We have strengthened wording around the important of the composition of zooplankton communities in response to a specific comment. However, we did not conduct primary analysis of grazers in this study, as the reviewer points out, multiple studies have examined Southern Ocean grazers in CMIP6.

During previous rounds of review, we have been strongly encouraged by reviewers to substantially reduce the amount of literature review this paper contains, so it is beyond the current scope to increase discussion points. However, we have created new links to zooplankton in relevant sections on phytoplankton community composition and signposted some of the suggested review papers which discuss grazers. In particular, Rohr et al. (2023) note that most CMIP models include a very limited representation of zooplankton (1-2) groups. Therefore, the species specific effect of zooplankton predation on phytoplankton which make them so important for ecosystem dynamics does not exist in the models. It is for this reason that we don't discuss grazing in the same depth as nutrients and light, because within the model architecture they are not capable of driving the population differences between phytoplankton types. We have strengthened this point in our manuscript adding:

"A lack of complexity in the representation of zooplankton communities accounts for a large degree of uncertainty in phytoplankton composition and the marine carbon cycle [*Rohr et al.*, 2023], this is particularly acute for the Southern Ocean where the major zooplankton fractions, salps and krill, feed on distinctly different size fractions of phytoplankton."

Additionally, we added a reference to *Heneghan et al.* [2023] who found improving the representation of phytoplankton communities would reduce uncertainty in Southern Ocean zooplankton projections, supporting our justification for focusing on phytoplankton in this manuscript.

3. **Definition of Southern Ocean:** The manuscript contains several definitions of the Southern Ocean. In the introduction, readers find that the Southern Ocean is south of the 30S (Gregg et al 2003); in Figure 1, it is south of 65S; in the spatial maps, all are south of 40S. And there are also SOOS regions. Why is this the case? Most messages are for the Antarctic Zone in the manuscript; why not only focus there or SOOS regions?

For most of the manuscript we use the region between 40-90°S as our working definition for the Southern Ocean, capturing the Transitional, Subantarctic and Antarctic zones per the precedent set by *Leung et al.* [2015] for investigating shifts between latitudinal bands in the Southern Ocean. They also include a subtropical zone between 30-40°S, which we reflect for example in Figure 5, but mapping out to 30°S for each of the parameters we asses would impair our ability to visualise spatial variability in the Antarctic zone. In the introduction we use 30°S as the boundary because the Southern Ocean basin area provided by *Gregg et al.* [2003] is calculated as being south of 30°S. The reviewer is correct that most changes occur in the Antarctic zone, south of 65°. Figure 1 is designed to be a "zoom in" on the Antarctic zone (65-90°S), being the region with the largest projected changes and the region most impacted by unrepresented processes in CMIP6 (e.g., ice shelf meltwater flux).

SOOS regions do not affect the definition of the Southern Ocean, and are superimposed on top of the maps to account for longitudinal spatial variability. This ensures that regional level variability is not lost by averaging across a latitudinal band which consists of both high

and low productivity. In a practical sense, assigning model values to an established observation framework can help to target additional observations to specific regions where the largest or most uncertain trends are projected.

4. **Conclusions:** There are some very vague statements, and some of them are not even supported by the analysis that are conducted in this manuscript.

We have made some changes to the conclusion as detailed in the specific comments.

Another critical point is the paper's title, "Biogeochemistry of Climate Driven Shifts in Southern Ocean Primary Producers." The main analysis is based on the results of the CMIP6 models, so it could be helpful to state this in the title. The methods section mentions, "Model and observational data for the Southern Ocean were collected and visualized." What kind of observational data did the authors collect? As seen in the manuscript, the only observational dataset used is the climatology from the World Ocean Atlas 2018. There is no detailed comparison between WOA data and the model results. In my opinion, It should be clearly communicated that this paper mainly aims to identify trends in CMIP6 models.

Yes, WOA2018 is the observational data set used.

We have added CMIP6 to the title. **"Climate Driven Shifts in Southern Ocean Primary Producers and Biogeochemistry in CMIP6 Models"**

Furthermore, all citations, including DOIs and formatting, should be checked, since some are missing.

**Missing citations in the text:** Nissen et al. (2021), Palter et al. (2010), Moore et al. (2018), Primeau et al. (2013) are listed in the references but need to be cited in the text. I did not check all the references, but there could be more examples of listed but not cited references in the manuscript.

These references were included in an earlier version of the manuscript and had been removed but not removed from the bibliography, this has been updated.

Overall, I think the current version of the manuscript requires significant revisions before it can be considered for publication.

**Specific comments**

**Abstract:**

**Line 26:**

Si* can be known by biogeochemists, but it would be good to specify it here.

Line 26: We have added an additional definition of Si* to the abstract "Si* ($[Si(OH)_4]$– $[NO_3^-]$)"

Misc is an abbreviation; writing it in a full-form would be better.

Misc changed to Miscellaneous

**Introduction:**

**Line 59:** It is difficult to get this information from Figure S1. Can you calculate and write the regions nearby? Adding a plot of zonal averages could be a good idea. Why is this SSP2-4.5 scenario, while all the other analyses are SSP5-8.5?

The proportion of productivity occurring within the Southern Ocean is the percentage of productivity which occurs south of 30°S vs the whole global ocean. 30°S is indicated on the y axis of Figure S1.

We have added this description to the figure caption.

Please see our response to the next comment re: producing zonal averages.

The value for Southern Ocean productivity using SSP2-4.5 is included because in a previous round of review we were asked to provide a more up to date figure for the percentage of global productivity which occurs in the Southern Ocean compared to the existing estimates in the literature which were ~10 years out of date. Creating a recent historic value is difficult in CMIP6 because the historic period only runs as far as 2015. Therefore, anything after 2015 is conducted as a future projection over a time period that has already passed, the SSP2-45 scenario most closely matches the real-world climate changes observed between 2015-2023, this is what we call a "middle of the road" scenario in the text. Everywhere else in the manuscript we use SSP5-85 as this reflects the reasonable worst-case scenario reached by 2100 which covers a substantial period of time where we are uncertain which trajectory we will follow.

Line 59: We have added a justification of the SSP2-4.5 scenario to the text "being the scenario mostly closely representing the true climate trajectory since 2015".

**Line 60:** I am also curious how much of this change comes from the Coast of Chile and West Africa

Developing a global scale assessment of regional productivity changes would be out of scope for this paper. We do not hold the regional masks necessary to conduct such an analysis outside of the Southern Ocean, which is the focus of this paper, it would be a massive amount of work to develop because it would require us to derive the basin area from the bathymetry for each region.

**Line 79:** To be more precise on the point about zooplankton, for example, authors already state the role of zooplankton in the Southern Ocean in the paragraph starting "In the sea-ice zone, grazing by zooplankton …." . This indicates the role of zooplankton in the region, but there is not much discussion about that later.

We have added more on this later (~Line 480) in response to another comment about zooplankton. The key issue around zooplankton in this manuscript is that while grazers are really important in controlling phytoplankton species composition in the Southern Ocean, the very basic parametrisation of them in CMIP6 means that the role of the grazing pressure is poorly represented. Therefore, the role of zooplankton is fairly minimal in determining drivers of phytoplankton composition in CMIP compared to temperature, light and nutrient availability. Hence, zooplankton are not discussed in as much detail as other drivers, however we do advocate for better representation of the grazing pressure and have expanded the conclusion to reinforce this point also.

**Line 100:** Henson et al. 2022 is a good reference for export fluxes, but phytoplankton losses through zooplankton grazing need other references.

Line 104: Added *Cavan et al.* [2017] as a more explicit reference to losses through zooplankton grazing.

**Line 115:** good reference but can be extended.

We have expanded the information on the *Swadling et al.* [2023] reference and extended to also include associated changes in ecosystem services with reductions in sea ice.

Line 118: "This is proposed to manifest in reduced algal, copepod, krill and fish populations with subsequent negative impacts on birds, mammals and associated ecosystem services [*Steiner et al.*, 2021]"

Line 120: 'Here we aim to … "It is difficult to follow whether authors reach these goals at the end of the manuscript. For me, a very clear example is the statement, "Here, we argue that targeting observations to establish phytoplankton-environment response interactions within the regions of the most rapid projected changes is essential to accelerate the improvement of phytoplankton representation in future generations of ESMs." I do not see this in the rest of the manuscript. If there is, can you make it more clear?

We removed the specific line quoted here. We do believe that we meet the objectives of the study, we have clarified this in the conclusion by addressing each aim in turn and stating the specific conclusions relating to each aim.

**Methods:**

**Table 1:** This table needs to be clarified. Did you use all the 'models selected' for each parameter? If yes, it means that, for example, you use six models for intppdiat but five models for limirrdiat. This example can be expanded. How does it affect your results? I see that in Figure 5, only six models that have intppdiat were used. Is this the case for other analyses? If yes, can you state it in your table?

Yes, all the models in the "Models selected" column were used to form the ensemble mean for the corresponding parameter, the selected models are based on those available for the specified parameter in the Pangeo catalogue.

The difference between the models for intppdiat and limirrdiat is because the CanESM5-CanOE model does not include the limirrdiat parameter. This point is explained on Line 145: "Only a small number of CMIP6 models contain irradiance limitation (*limirr*) and iron limitation (*limfe*) for multiple phytoplankton types, therefore analyses of light and iron limitation of phytoplankton utilise <5 models".

This should not affect any results because we do not make direct (statistical) comparisons between ensembles containing different model members. For example, where iron limitation of diatoms and growth are correlated (Figure 4) this is done on a per model basis. And where diatom productivity (intppdiat) is directly compared to total productivity (intpp) we excluded models which did not include intppdiat from the intpp ensemble, as explained on Line 140: "Where an analysis type relied on the direct comparison between two or more parameters, only models that contained both parameters were selected."

The table caption has been expanded to reinforce this point.

"Table 1: Selected models used in analysis of CMIP6 data based on availability in the Pangeo catalouge. Models shown in bold represent multiple resolutions of the same core model which were subsequently averaged prior to calculation of the ensemble mean. Where direct comparisons are made between multiple parameters, ensembles were adjusted to include only those models which were existed for all of the selected parameters."

**2.2 Regional data**

I cannot easily connect the analysis of SOOS with the rest of the manuscript. Why is the manuscript suddenly analyzing only Si*, pH, and temperature but not nutrient limitation, iron, NO3, phytoplankton classes, etc. for these subregions? Also, the main focus was always NPP and trying to link it with nutrient limitation, light, and MLD, but suddenly, there is a chlorophyll figure (Figure

3). What is the purpose? If it is about Chl:C ratios, the figure for the carbon fields of phytoplankton can be useful.

The SOOS section is about adding capacity for observational efforts, a large part of this is to do with remote sensing of variables. Therefore, for the regional analysis we focus on the variables which are easiest to determine from platforms such as argo floats and satellites. For that reason, we focus on chlorophyll over carbon in this section, because it is far easier to measure. These should be seen as the incorporation of additional commonly measured variables, having not previously presented temperature, pH or nutrient data in the manuscript, rather than a narrowing of focus on specific variables for the SOOS section. We are not seeking to comment on Chl:C ratios in the manuscript.

We have added a justification to the methods section:

Lines 174-178: "Commonly observed marine variables (chlorophyll, temperature, nutrients, pH) were analysed by their regional sector using the Southern Ocean Observing System (SOSS) framework. The purpose of integrating CMIP6 projections within the main existing observation framework is to identify regions of the largest expected changes in routinely observed variables to inform the coordination of future sampling efforts by the SOOS regional working groups."

**Results and Discussion:**

**3.1 Physical climate drives biological changes in Southern Ocean water masses**

**Line 177:** "increased buoyancy fluxes (including freshwater inputs) act as opposing drivers of stratification," can you show buoyancy fluxes as supplementary figures? It is visible that other factors determine the MLD when Figure 2A and Figure 2B are compared. However, it would be better to see them.

This would be a really interesting parameter to include however, it is not possible to resolve the freshwater flux from CMIP6 as, because explained previously in this response, a major weakness of CMIP6 models is the lack of ice shelf representation. Therefore, there are no parameters within CMIP6 which would allow us to derive ice melt or the subsequent meltwater flux which contribute towards buoyancy fluxes. The model representation is limited to the final MLD change, but we recognise that a number of physical processes contribute towards the spatially heterogenous changes in MLD. We identify changes in stratification as being one of the major targets for improvement in future generations of CMIP for this reason.

**Line 196:** "CMIP6 models project the greatest increase in productivity to occur across the coastal zone of the Southern Ocean (65-90°S) (Figure 2c), where irradiance limitation is reduced (Figure 2d)." Is this really the case? When I look at Figure 2C, the most significant increase ('dark red areas') is more in the transitional zones. Ideally, it would be nice to see some numbers to get an idea of the magnitude of the increase.

Figure 2C is showing absolute change in productivity, but the "greatest increase" is really about relative change. I.e., although the change (in gC) is similar between the Antarctic and Transitional zones, the starting value of productivity in the Antarctic is much lower, so the relative increase is orders of magnitude greater in the Antarctic compared to Transitional zone. We have clarified that this is relative change in the text and added a new supplementary figure showing % change to support this point.

[Figure]

**Figure S3: Percentage change in average annual vertically integrated primary productivity across the Southern Ocean resulting from diatoms for 2090-2100 under SSP5-8.5 compared to a historical mean (1985-2015).** Representative of a multi-model ensemble of CMIP6 models; models included are detailed in Table 1.

**Line 197:** "Conversely, across the Transitional zone (40-50°S), ….. " It would be good to discuss why IPAR is changing. For example, IPAR is increasing only close to Antarctica, but the limitation reduction occurs in the larger area; why is this the case? Is it related to parametrizations of light limitation in the models?  Is it about clouds or water column structure?

In the transitional zone, clouds are most likely to be playing the major role in increasing light limitation because IPAR is predominantly driven by clouds and sea ice cover in CMIP6, this is consistent with the reductions in IPAR.

However, the reviewer makes an interesting point about the Antarctic zone where sea ice changes only lead to increases in IPAR very close to the Antarctic continent, yet light limitation reduces across a much wider area. The explanation for this is not actually related to light, but the co-current increase in iron limitation across this region. As productivity increases, nutrient utilisation increases and a greater proportion of the phytoplankton are hitting iron limitation before they reach light limitation so iron limitation increases and light limitation decreases.

We have added an explanation of this:

Line 206: "…IPAR reduces (Figure 2e) with cloud cover driving changes in light beyond the sea ice zone".

Line 211-213 "Increased iron limitation across much of the Antarctic and Subantarctic coincides with reduced light limitation (Figure 2d) beyond the region where IPAR increases (Figure 2e), this is

consistent with greater iron demand from increased productivity (Figure 2c) driving phytoplankton to iron limitation before light limitation. "

**3.2 Changing biogeochemistry of the Southern Ocean**

**3.2.1 Micronutrient supply and uptake**

**Line 227:** Iron limitation is expected to increase most in the transitional zone …." I understand that the change in the wind can affect the atmospheric deposition, and if it is shown that the wind changes, it will be affected in the future. Do these models have atmospheric iron deposition, and can you talk about it?  Are there differences among the models?

The models do not include atmospheric deposition, just iron concentration and iron limitation. There is a parameter for surface run off (i.e., riverine input from sediment dissolution) but nothing for atmospheric deposition. We already discuss the role of atmospheric deposition, for example the fact that the flux is low compared to other iron sources. Line 237: "supply limitation from low atmospheric inputs and significant distances from terrigenous sources". We place the changing iron limitation in the context of wind speed changes for the transitional zone, stating "Iron limitation is expected to increase most in the transitional zone between South America and New Zealand, correlating with a reduction in near-surface wind speed (Figure 2a), suggesting that atmospheric deposition of iron will decline in this region" (Line 243). However, this is based on correlation and balance of probability, with the transitional zone being far away from land in deep waters where sediment resuspension is low and resupply through upwelling would be expected to be low. It is not possible to put direct values on this as being a result of declines in deposition because of the lack of parameterisation.

**Line 238:** "At the group level, iron limitation…" needs a reference,

Added references to [*Timmermans et al.*, 2004] and [*Hudson and Morel*, 1990]

**3.2.2 Macronutrient supply and uptake**

**Line 256:** "Although projections indicate an increase in chlorophyll across such regions (Figure 3), models do not show any increases in nitrate limitation." is it really 'any change'? Figure S3 shows some change until +-20%, which seems higher than the iron limitation.

The "regions" referred to are across the coastal zone (see prior line), and the point of this sentence is to say that although parts of the coastal region experience periodic nitrate limitation during blooms, there is no widespread modelled change to the existing nitrate replete environment expected. The regions the reviewer refers to as having 20% changes in nitrate limitation are the northernmost parts of the Subtropics which represent a much more oligotrophic environment compared to the majority of the Southern Ocean.

Line 268: We have clarified the wording here. Changed to "models do not show any substantial increases in nitrate limitation south of the Subtropical zone".

**Line 271:** "Changes in Si* correlate with increases in chlorophyll concentration across the same regions (Figure 3), indicative of increased phytoplankton concentrations resulting in a drawdown of silicic acid." To see this, can you plot the Si* as a subplot and provide the reader with some numbers (e.g. correlation coefficient)? Ideally, comparing models and WOA in maps is a good idea. This could also support the analysis in Table 2.

The change in Si* is already plotted in Figure S9. Statistically, it wouldn't be possible to perform a linear regression analysis between the SI* and Chlorophyll maps because of the fact the pixels themselves are not uncorrelated. Due to the interpolations performed, the value of one pixel affects the neighbouring pixel so they are not independent data points and therefore the two

datasets do not fulfil the statistical requirements for correlation analyses. You could take a bulk approach to this, i.e., creating one correlation value for each of the SOOS zones, but because of the heterogeneity of both SI* and Chlorophyll within the zones, it was not an informative metric when we tried this since the uncertainty was large.

On reflection "correlate" was not the best word to use in this paragraph, so this has been changed to "Decreases in Si* coincide with increases in chlorophyll concentration across the same regions (Figure 3), concurring with increased phytoplankton concentrations resulting in a drawdown of silicic acid".(Line 281)

**Figure 3:** Just a curiosity, Why did you choose to 500 m to integrate chlorophyll?

It's an over cautious depth to include all chlorophyll. Different models use different depth intervals and 500m was one of the ubiquitous depths that all models include which makes it easier to apply a consistent integration function.

**3.3 Primary production and representation in CMIP6**

**3.3.1 Phytoplankton classes**

To follow the numbers in Figure 4 is difficult. 55%, 74 % etc. Is the second y-label missing in figures A,B,C,D,E,F,? I cannot see these numbers anywhere. In general, Figure 4 should be better described in the related section.

Figure 4 was updated in a previous round of review and we omitted to update the values in the text, these should be read offs of the slope value for the relevant figures, these have now been updated.

There is not meant to be a second y-label on the figures, we have grouped them where possible. E.g. A,B,D,E all have total primary productivity as the X label. We have moved the Y label to the left side for D,E,F to make this consistent with the other sub plots and hopefully easier to interpret.

**3.3.2 Ecological dynamics and ecophysiology**

This subsection does not really say about ecological dynamics or ecophysiology. If this is the aim, It needs to be extended. If not, it can be combined with another subsection. In this subsection, it is only referred to Figure 4. If the authors want to keep it, I suggest that zooplankton-related discussion be added to this subsection as well.

We have merged this with section 3.3.1 (now 3.3).

**Line 379:** 'In a changing ocean,>…. ' needs more references

Added an additional reference:

Line 376: "In a changing ocean, phytoplankton will succeed where they have the greatest biological plasticity, for example the ability to photo-acclimate rapidly [*Arrigo et al.*, 2010] or utilise a diverse range of nutrients [*Kwon et al.*, 2022]."

**Line 387:** if it is written 'little correlation', please provide numbers.

Line 385: Added: "with an $R^2$ of 0.51 for picphytoplankton and 0.06 for diatoms (Figure 4 g,h)".

**3.4 Latitudinal productivity projections CMIP6.**

Dot '.' In the section title is not need

Removed

**Line 415:** "however ….. " Why is this the case? Is this related to temperature or nutrient limitation parametrization? I suggest to discuss it. In addition, I guess that the unit in Figure S7's y-axis should be %.

Yes the axis was labelled incorrectly here, this has been corrected to %.

Line 430: Added "Compositional changes in primary productivity across the subtropics are driven by reductions in nutrient availability from increased stratification [*Fu et al.*, 2016] , as reflected in the increase in nitrate limitation across these regions (Figure S4). Under nutrient stress smaller phytoplankton can outperform larger phytoplankton, here diatoms, because of lower nutrient demands, leading to the modelled declines in diatom populations (Figure 7). The divergence in nutrient half saturation constants between different models (see 3.3) subsequently leads to a wide range of projections in species composition when nutrients become limiting."

**Figure 5:** I guess that Figure 5 shows the spatial average of latitudinal bands. I suggest specifying it in the caption. In addition, I suggest showing the spatial maps of change in total, diatom, and non-diatom NPP change in the supporting information (like Figure 2) to support Figure 5.

We have revised the caption to specify that it represents the spatial average. We have added two new supplementary figures to show the spatial change in diatom and non-diatom NPP (shown below).

[Figure]

**Figure S5: Anomaly in average annual vertically integrated primary productivity across the Southern Ocean resulting from diatoms for 2090-2100 under SSP5-8.5 compared to a historical mean (1985-2015).** Representative of a multi-model ensemble of CMIP6 models; models included are detailed in Table 1

[Figure]

**Figure S6: Anomaly in average annual vertically integrated primary productivity across the Southern Ocean resulting from non-diatoms (intpp-intppdiat) for 2090-2100 under SSP5-8.5 compared to a historical mean (1985-2015).** Representative of a multi-model ensemble of CMIP6 models; models included are detailed in Table 1

**4.Conclusions**

**Line 463**: AIS was not mentioned before, write full form before using abbreviation

Antarctic Ice Sheet definition added

**Line 464**: "the current iteration …. " what about Antarctic zone? It contradicts with what is written in Line 276 "Therefore, the large chlorophyll increase and large Si* decline projected in the Weddell Sea is likely driven by an increase in diatoms, whereas the productivity increase with only small changes in both chlorophyll and Si* seen on the west Antarctic Peninsula probably results from an expansion of non-diatom phytoplankton with lower Chl:C."

These two lines are talking about different spatial scales of change. Line 464 is referring to the fact that there are no significant changes evident in species composition across the latitudinal bands, besides a decrease in diatoms in the subtropics. Line 276 is discussing longitudinal differences within the Antarctic zone. These are consistent because if there is a shift towards diatoms in the Weddell Sea and a shift towards non-diatoms along the WAP, then the two would average out to result in no substantial change across the Antarctic zone as a whole.

The new supplementary figures the reviewer suggested we add supports this theory, we have adapted the text to refer to these new figures and make the distinction in scales clearer.

Line 467 revised to: "The current iteration of CMIP6 models do not suggest any significant shifts in the average community composition between latitudinal bands across the Southern Ocean, outside of a decrease in the relative abundance of diatoms in the subtropical zone."

Line 286 revised to: "Therefore, the large chlorophyll increase and large Si* decline projected in the Weddell Sea are driven by an increase in diatoms (Figure S5), whereas the productivity

increase with only small changes in both chlorophyll and Si* seen on the west Antarctic Peninsula result from an expansion of non-diatom phytoplankton with lower Chl:C (Figure S6)."

**Line 465**: "However, there is ....." is it really the case all models do not represent grazers. If yes, can you specify and discuss in the discussion part the possible effect of it? You can check Rohr et al. (2023) for this purpose. In addition, where readers can see large uncertainty of up to ±30% ?

The specific point on this line is about the lack of inclusion for grazer community composition: "key processes which will impact phytoplankton (e.g., viral losses, **composition of the grazer community**) are absent from most models"

Models contain a grazer term but often do not distinguish between, for example, krill and salps which consume different fractions of the phytoplankton pool and thus lead to significant uncertainties in phytoplankton composition in the Southern Ocean.

This reinforces the paper by Rohr et al., (2023) which says "coupled-climate models that typically include only 1–2 groups. This leads to substantial uncertainty in how modelers implicitly represent complex communities and their impact on marine carbon cycling."

Line 470: Added "A lack of complexity in the representation of zooplankton communities accounts for a large degree of uncertainty in phytoplankton composition and the marine carbon cycle [*Rohr et al.*, 2023], this is particularly acute for the Southern Ocean where the major zooplankton fractions, salps and krill, feed on distinctly different size fractions of phytoplankton."

Figure S10 added as the reference for the 30% uncertainty figure.

**Line 477**: If "literature contains wide divergence...", you need to cite them.

This is a conclusionary reference back to an earlier part of the text where the relevant papers are cited and discussed. Added a reference to the earlier section "(see 3.3)".

**4.2 Observational recommendations:**

**Line 482:** Double dot ':" is not needed in the subsection title

Removed

**Line 483:** "We have identified changes in nutrient upwelling ....." Where did you identify the nutrient upwelling strength, eddy strengthening, mutualism and resource competition in the manuscript?

This sentence was removed from the manuscript

**Line 508:** "A focus towards defining variability in phytoplankton buoyancy, carbon uptake and release as dissolved organic matter, grazing by zooplankton, interactions with viruses, macronutrient stoichiometry and micronutrient utilisation will enhance our core understanding of Southern Ocean phytoplankton dynamics in a changing climate." I am having difficulty understanding this statement. There is existing literature (modeling, observational, experimental) that relates to the points made here. Could you clarify the intended message of this sentence?

This sentence is no longer part of the revised conclusion.

**Response to Reviewer 2**

Reviewer comments in black, author responses in blue

This is the second time authors submit this paper, and I recognize authors have made a great effort to improve the manuscript from the previous round. I acknowledge authors kept most of the well-written introduction, while improving, and shorten and simplify, the results and discussion sections. My main criticism of the old version was that the original results of the study were diluted in a (otherwise) good review of the state of the art literature, thus obscuring the contributions of the paper. In this version, the analyses are more clearly linked to existing literature, although I still think that most of the information gathered from the study seems mostly a review. Sometimes it is difficult to discern what is an original result from literature.

I also acknowledge the rearrangement of Table 1, though I think it can be presented as supplementary material.

Although the paper is focused on the biogeochemistry, I believe that some discussion on the effect of grazing to primary producers would be beneficial to complement the effects of iron and light limitations.

I found some sentences difficult to follow (the long sentence in lines 367 to 371 is a good example). I would like authors to revisit them for improving readability.

Finally, I think that the last section should be revisited. Some sentences point out to processes that, to my understanding, are not presented in the present study. It seems that recommendations are based to general statements, but not explicitly to the results found in the study.

In summary, I think the paper has benefited from the changes made in the previous iteration, but some additional changes need to be considered before publication.

We thank the reviewer for their further comments on our manuscript and appreciate their comments on the progress made from the previous version.

**Re Table 1**: It is a large table, but forms the bulk of the methods because it contains the model details, these are quite important for interpreting the figures in knowing how large the relative ensembles are. It was originally a supplementary table but we were asked to bring it in to the main text by another reviewer. The editor has now confirmed this should remain part of the main text.

**Grazers**: We recognise that grazers are a crucial part of the Southern Ocean ecosystem, not least as a driver of phytoplankton community composition. Multiple papers have synthesised grazer populations in CMIP6, so we did not select zooplankton as a primary variable to analyse in this study. In particular, *Rohr et al.* [2023] note that most CMIP models include a very limited representation of zooplankton (1-2) groups. Therefore, the species specific effect of zooplankton predation on phytoplankton which make them so important for ecosystem dynamics does not exist in the models. It is for this reason that we don't discuss grazing in the same depth as nutrients and light, because within the model architecture they are not capable of driving the population differences between phytoplankton types. We have strengthened this point in our manuscript adding:

"A lack of complexity in the representation of zooplankton communities accounts for a large degree of uncertainty in phytoplankton composition and the marine carbon cycle [*Rohr et al.*, 2023], this is particularly acute for the Southern Ocean where the major zooplankton fractions, salps and krill, feed on distinctly different size fractions of phytoplankton."

We have added further mentions of the role of grazers as potential drivers of phytoplankton species composition throughout, including reference to the relevant CMIP6 zooplankton synthesis paper [*Rohr et al.*, 2023].

Additionally, we added a reference to *Heneghan et al.* [2023] who found improving the representation of phytoplankton communities would reduce uncertainty in Southern Ocean zooplankton projections, supporting our justification for focusing on phytoplankton in this manuscript.

**Sentence Length:** We have revisited long sentences in the manuscript and restructured these to be more succinct, including the sentence identified by the reviewer.

**Conclusion**: We have removed the previous end part of the conclusion which referred to multiple processes not directly covered in the study and refocused this on the specific recommendations arising from our results.

**Specific comments about the text and figures:**

Line 45: I think this sentence can be extended up to the global ocean (e.g., Jiang et al., 2019).

Line 45: Changed to: "As the Ocean's ability to buffer increased concentrations of atmospheric $CO_2$ weakens [*Jiang et al.*, 2019], the role of pelagic ecosystems are expected to become increasingly important in Southern Ocean carbon uptake (Henley et al., 2020)."

Lines 57-58: "...the Southern Ocean phytoplankton…represented xxx% of global marine net primary productivity, which is equivalent to…".

Line 59-60: Changed to: "Southern Ocean phytoplankton (defined as those south of 30°S per Gregg et al. (2003)) represented 36.31% of global marine net primary productivity, equivalent to 15.5 Pg C $yr^{-1}$ (Figure S1)".

Line 169: I think a typo here: "silicic acid plus sea surface temperature".

Corrected

Line 170: Si* is introduced in line 263. Please move it to here.

Si* definition moved; it is also now in the abstract as a formulaic expression.

Line 198: What about the increase in PP in lower latitudes; and far from the antarctic coast? In fact, these latitudes are still the Southern Ocean? It is not clear what the limits of the Southern Ocean are considered according to the text.

We have reworded this to mean relative change, the absolute change in productivity is similar between the coast and lower latitudes but the relative change in far higher in the coast. We added a new supplementary figure to show this (Figure S3). We have clarified Southern Ocean spatial definitions at the relevant points in the text, in general we consider the Southern Ocean to be south of 30°S because this is consistent with previous zonal assessments of the Southern Ocean in CMIP e.g. [*Leung et al.*, 2015], but we mostly focus on the region south of 40°S for plotting to aid visualisation of the coastal Antarctic zone, where the largest changes occur.

Lines 204-209: "CMIP6 models project the greatest relative increase in productivity to occur across the Antarctic zone of the Southern Ocean (65-90°S) (Figure 2c, Figure S3), where irradiance limitation is reduced (Figure 2d). Conversely, across the Subtropical zone (50-65°S), IPAR reduces (Figure 2e) with cloud cover driving changes in light beyond the sea ice zone, irradiance limitation increases (Figure 2e) and productivity changes are lesser here compared to the coastal Southern Ocean (Figure 2c, Figure S3)."

[Figure]

**Figure S3: Percentage change in average annual vertically integrated primary productivity across the Southern Ocean resulting from diatoms for 2090-2100 under SSP5-8.5 compared to a historical mean (1985-2015).** Representative of a multi-model ensemble of CMIP6 models; models included are detailed in Table 1.

Lines 201 to 205: Are these changes a result of the study, or are referred to other studies?

"reduced upper-ocean stratification from strengthening zonal winds" comes from this study, but are also observed in the referenced studies. We have added a Figure reference to 2a (strengthening zonal winds) and 2b (reduced upper ocean stratification) to clarify this.

The changes to iron supply do not directly come from this study, because mechanisms of supply are not included in the models, but we discuss this as a potential implication of changes in stratification, which are a result of this study.

Lines 207 to 210: Please rephrase this sentence; it can be separated into 3 sentences. Moreover, where are the Ross and Flicher-Ronne ice shelves locations indicated?

We have rephrased this section and added locations of the ice shelves.

Lines 219-223: "Models generally agree on changes in summertime mixed layer depth across most of the open ocean (Figure S2b), the greatest source of uncertainty is at the terminus of the Ross (Ross Sea) and Flicher-Ronne (Weddell Sea) ice shelves. This uncertainty in stratification can be linked to the lack of representation of ice shelves and their meltwater flux in the current generation of CMIP models (Purich and England, 2021)."

Figure 2: I would include an axis of the latitude to get a sense when detailed in the text.

We have added the latitudinal bounds to Figure 2a to indicate the zones without obscuring all of the subpanels. The figure caption has been updated to reflect this.

Lines 236 to 247: This seems more of a review.

We have shortened this section, we felt it was important to acknowledge the role of micronutrients beyond what is currently represented in the models as this is currently an active area of research. We felt it could be useful to place ongoing observational and experimental work within the context of future model development.

Line 271 and others: Sometimes sentences are awkwardly written as this one; "...across the same regions, which is indicative of…".

We have rewritten this sentence to

Lines 281-283: "Decreases in Si* coincide with increases in chlorophyll concentration across the same regions (Figure 3), concurring with increased phytoplankton concentrations resulting in a drawdown of silicic acid."

Line 292: "there is no evidence for a direct effect of acidification on phytoplankton…". Please review this long sentence.

Merged with the previous sentence

Lines 300-302: "Projected changes in pH do not differ regionally, and show little variation within regions (low standard deviation), therefore regional changes in phytoplankton growth do not result from a direct acidification effect"

Line 300: Though interesting, I don't know if this paragraph is necessary as it does not present or discuss any direct result from the paper.

The point of this paragraph is to acknowledge that while our results do not show a direct acidification effect which impacts phytoplankton, the interactive effect of pH on stoichiometry could impact phytoplankton if we are to increase model complexity. We have changed this paragraph to make the reference to our pH results clearer, and reduced the length of the literature discussion.

Line 359: What authors referred by "based on species"?

Changed to: "for different diatom species".

Lines 379 to 388: May the role of grazing be discussed also here.

This paragraph didn't seem like the best place to expand the discussion on grazing because this section specifically focuses on ecophysiology and the difference in growth response to iron and light limitation in two specific models. However, this section was merged in to the wider 3.3 on Primary Productivity, to which some additional discussion on grazing was incorporated in the context of developing models to include specific phytoplankton-zooplankton predation patterns.

Line 448: A citation is lacking here.

Added (Figure 5d) as the reference for this.

Section 4.1. Are these really "Implications"? Results are summarized and suggestions are provided.

This subheading was removed, it is now just conclusions.

Line 484 to 486: I don't know whether this sentence applies to the manuscript results. Where changes in nutrient upwellling, mutualism, resource competition, etc, have been identified in the text?

We have removed this sentence from the manuscript.

Line 491 to 495: Please reformulate this sentence.

We have split this sentence:

"The abundance of diatoms between 30°S and 65°S represents one of the greatest sources of uncertainty in phytoplankton community composition. This could be related to poor model representation of diatom species in these regions due to data sparsity, as sampling of open waters is more seasonally limited in comparison to land-based stations in the Antarctic zone, where there is a lower uncertainty for diatom abundance."

Additional references:

Arrigo, K. R., M. M. Mills, L. R. Kropuenske, G. L. van Dijken, A. C. Alderkamp, and D. H. Robinson (2010), Photophysiology in two major southern ocean phytoplankton taxa: photosynthesis and growth of Phaeocystis antarctica and Fragilariopsis cylindrus under different irradiance levels, *Integr Comp Biol*, *50*(6), 950-966.

Cavan, E. L., S. A. Henson, A. Belcher, and R. Sanders (2017), Role of zooplankton in determining the efficiency of the biological carbon pump, *Biogeosciences*, *14*(1), 177-186.

Death, R., J. L. Wadham, F. Monteiro, A. M. Le Brocq, M. Tranter, A. Ridgwell, S. Dutkiewicz, and R. Raiswell (2014), Antarctic ice sheet fertilises the Southern Ocean, *Biogeosciences*, *11*(10), 2635-2643.

Fu, W. W., J. T. Randerson, and J. K. Moore (2016), Climate change impacts on net primary production (NPP) and export production (EP) regulated by increasing stratification and phytoplankton community structure in the CMIP5 models, *Biogeosciences*, *13*(18), 5151-5170.

Gregg, W. W., M. E. Conkright, P. Ginoux, J. E. O'Reilly, and N. W. Casey (2003), Ocean primary production and climate: Global decadal changes, *Geophysical Research Letters*, *30*(15).

Heneghan, R. F., J. D. Everett, J. L. Blanchard, P. Sykes, and A. J. Richardson (2023), Climate-driven zooplankton shifts cause large-scale declines in food quality for fish, *Nature Climate Change*, *13*(5), 470-477.

Hudson, R. J., and F. M. Morel (1990), Iron transport in marine phytoplankton: Kinetics of cellular and medium coordination reactions, *Limnology and Oceanography*, *35*(5), 1002-1020.

Jiang, L.-Q., B. R. Carter, R. A. Feely, S. K. Lauvset, and A. Olsen (2019), Surface ocean pH and buffer capacity: past, present and future, *Sci Rep*, *9*(1), 18624.

Kwon, E. Y., M. G. Sreeush, A. Timmermann, D. M. Karl, M. J. Church, S.-S. Lee, and R. Yamaguchi (2022), Nutrient uptake plasticity in phytoplankton sustains future ocean net primary production, *Science Advances*, *8*(51), eadd2475.

Leung, S., A. Cabré, and I. Marinov (2015), A latitudinally banded phytoplankton response to 21st century climate change in the Southern Ocean across the CMIP5 model suite, *Biogeosciences*, *12*(19), 5715-5734.

Purich, A., and M. H. England (2021), Historical and Future Projected Warming of Antarctic Shelf Bottom Water in CMIP6 Models, *Geophysical Research Letters*, *48*(10), e2021GL092752.

Rohr, T., A. J. Richardson, A. Lenton, M. A. Chamberlain, and E. H. Shadwick (2023), Zooplankton grazing is the largest source of uncertainty for marine carbon cycling in CMIP6 models, *Communications Earth & Environment*, *4*(1), 212.

Steiner, N. S., J. Bowman, K. Campbell, M. Chierici, E. Eronen-Rasimus, M. Falardeau, H. Flores, A. Fransson, H. Herr, and S. J. Insley (2021), Climate change impacts on sea-ice ecosystems and associated ecosystem services, *Elem Sci Anth*, *9*(1), 00007.

Swadling, K. M., et al. (2023), Biological responses to change in Antarctic sea ice habitats, *Frontiers in Ecology and Evolution*, *10*.

Timmermans, K. R., B. van der Wagt, and H. J. W. de Baar (2004), Growth rates, half-saturation constants, and silicate, nitrate, and phosphate depletion in relation to iron availability of four large, open-ocean diatoms from the Southern Ocean, *Limnology and Oceanography*, *49*(6), 2141-2151.

---

## Referee Report (RR1)

Review of the manuscript « Climate driven shifts in Southern Ocean primary producers and biogeochemistry in CMIP6 models».

This is the third time authors submit this paper. I again acknowledge that most of the introduction stays. Results and discussion section has been further simplified, which is something I acknowledge as well. I criticized the old versions mostly because I think that original results presented in the text were diluted within a review, instead of highlighting the relevance of the present work. I am happy to say that this version further improves this issue.

I still think that Table 1 can be presented as supplementary material.

I acknowledge as well the inclusion of some discussion on the effect of zooplankton grazing.

I think the paper has benefited from the changes made in the previous two iterations.

Specific comments about the text and figures:

Line 46-47: "As the ocean's buffering capacity increases, atmospheric CO2 concentrations weaken (Jiang et al., 2019), and the role of pelagic ecosystems is expected to become more important in the Southern Ocean's carbon uptake. (Henley et al., 2020)".
Line 58: I will remove "across the Southern Ocean", as it is not the only place were phyto fuels ecosystems.
Line 60: I would say something like: "which is the scenario that comes closest to representing the current climate trajectory".
Line 96-100: I think this can be removed.
Line 125: we?
Line 184: May acronyms be defined for these parameters.
Line 275: "...where iron-manganese…" can be removed, and "yet only iron…" joined with precedent sentence.
Line 334: I think authors mean that changes in phytoplankton growth don't result from acidification in CMIP6 results, as they explain after how these effects have impacts.
Line 390: Why these numbers have been corrected?

---

## Author Response (AR2)

Climate driven shifts in Southern Ocean primary producers and biogeochemistry in CMIP6 models.

Fisher et al.,

Review Response: Round 2

**Reviewer 1**

I acknowledge the authors' efforts in addressing the first round of revisions. The manuscript is now more understandable, and the connection between the figures and the text is clearer. While I don't have many specific suggestions (except for the two listed below), I believe there is still room for improvement in the linkage between the figures and the text. This may be because much of the CMIP6-related analysis provided by the authors is either buried within the discussion of existing literature or lacks direct support, as some diagnostics (e.g., freshwater flux) do not have output from the CMIP6 models.

We thank the reviewer for their comments. As the reviewer acknowledges, a key struggle is that some parameters are missing from CMIP6 models, we have made an effort to be explicit about which parameters are absent from the models and hope this paper can be used as a basis for increasing the representation of biogeochemically important processes in future generations of CMIP.

I have two additional suggestions/corrections:

1. Figure 1 is not fully described in the manuscript. It is referred only in line 67 when the authors says which change is unknown. A more detailed caption would help the reader. The authors attempt to link Figure 1 with Table S1, but either the references are incomplete or some values (e.g., pH) do not match in table and figure, or certain values (like IPAR) are missing in the table. It is important to clarify where these numbers are coming from, as it is difficult to follow. For example, pco2 row "pCO2 +200% Increase from ~500 µatm (GLODAP) to ~1000 µatm under RCP8.5" Isn't it the increase is 100%? Also if GLODAP is referred, it is needed to be in the references.

We have expanded the figure caption to make clear which parameters in Figure 1 result from CMIP6 values derived in this study compared to those which come from literature values.

   **"Figure 1: Schematic diagram of Southern Ocean pressures associated with climate change and the downstream biogeochemical consequences for ecosystem productivity.** Values shown for surface warming, surface insolation and pH are 100 year mean changes to 2100 under the SSP5-8.5 scenario south of 65°S and are taken from CMIP6 models. Literature values are used for changes in $pCO_2$, stratification, and shelf warming at depth for the same time period and climate trajectory (Kawaguchi et al., 2013;Hauck et al., 2015;Purich and England, 2021)  (see Table S1 for a full description). Question marks indicate key processes which drive biogeochemical change but are not currently included in CMIP models and therefore estimations of change do not currently exist."

We have made clearer reference to Figure 1 in the text, adding in another reference to this Figure in reference to IPAR.

Line 67: "This increase in ocean surface area available for light transmission is offset by decreased insolation (-8.3%, Figure 1) associated with a greater degree of cloud cover."

We have updated Figure 1 to correct the $pCO_2$ change to 100%

We are not directly referencing GLODAP data, but we are obtaining the $pCO_2$ value from Kawaguchi et al., (2013) which performs an analysis of GLODAP data. We have added the Kawaguchi et al (2013) reference to the Figure 1 caption.

We have checked Table S1 to make sure the values are consistent with the Figure 1 and added a description of the IPAR data to this.

2. Figure 2A: The label for Subantarctic zone is incorrect; it is written as "Subtropical." This needs to be corrected.

Thank you, we have corrected this label.

**Reviewer 2**

This is the third time authors submit this paper. I again acknowledge that most of the introduction stays. Results and discussion section has been further simplified, which is something I acknowledge as well. I criticized the old versions mostly because I think that original results presented in the text were diluted within a review, instead of highlighting the relevance of the present work. I am happy to say that this version further improves this issue. I still think that Table 1 can be presented as supplementary material. I acknowledge as well the inclusion of some discussion on the effect of zooplankton grazing. I think the paper has benefited from the changes made in the previous two iterations.

We thank the reviewer for the comments on improvements we have made to the paper. Regarding Table 1, the editorial opinion was to leave this in the main text.

Specific comments about the text and figures:

 Line 46-47: "As the ocean's buffering capacity increases, atmospheric CO2 concentrations weaken (Jiang et al., 2019), and the role of pelagic ecosystems is expected to become more important in the Southern Ocean's carbon uptake. (Henley et al., 2020)".

This isn't correct, the point is that the buffering capacity of the ocean is decreasing not increasing. We have changed the original statement to be clearer

"As the Ocean's buffering capacity for increasing concentrations of atmospheric $CO_2$ reduces…."

Line 58: I will remove "across the Southern Ocean", as it is not the only place were phyto fuels ecosystems.

Removed

Line 60: I would say something like: "which is the scenario that comes closest to representing the current climate trajectory".

Changed

Line 96-100: I think this can be removed.

This sentence refers to the fact that surface chlorophyll representation has not improved between CMIP5 and CMIP6, based on benchmarking studies. Given that our paper is arguing for improved representation of biogeochemical processes linked to productivity, we feel this is an essential part of the introduction, providing the rationale for the study which should be retained.

Line 125: we?

Yes, we have used we consistently throughout the manuscript.

Line 184: May acronyms be defined for these parameters.

It is unclear what this comment is referring to, there are no acronyms on line 184 or even in the subsequent paragraph. All acronyms are defined in the text and Table 1 for the parameters.

Line 275: "…where iron-manganese…" can be removed, and "yet only iron…" joined with precedent sentence.

Changed

 Line 334: I think authors mean that changes in phytoplankton growth don't result from acidification in CMIP6 results, as they explain after how these effects have impacts.

Correct, we are saying the phytoplankton growth doesn't result from acidification in CMIP6.

Line 390: Why these numbers have been corrected?

There was a minor piece of reanalysis performed in an early round of revisions to change the y axis units from seconds to days, removing exponentials, which changed a couple of the slope values by ~2% points, but this had not been carried forward in to the text. This revision ensured consistency between the figures and the text, the minor change in numbers does not impact any of the analysis or conclusions.